# Cationic amino acid transporters play key roles in the survival and transmission of apicomplexan parasites

Esther Rajendran[1], Sanduni V. Hapuarachchi[1], Catherine M. Miller[2], Stephen J. Fairweather[1], Yeping Cai[3], Nicholas C. Smith[4], Ian A. Cockburn[3], Stefan Bröer[1], Kiaran Kirk[1] & Giel G. van Dooren[1]

Apicomplexans are obligate intracellular parasites that scavenge essential nutrients from their hosts via transporter proteins on their plasma membrane. The identities of the transporters that mediate amino acid uptake into apicomplexans are unknown. Here we demonstrate that members of an apicomplexan-specific protein family—the Novel Putative Transporters (NPTs)—play key roles in the uptake of cationic amino acids. We show that an NPT from *Toxoplasma gondii* (*Tg*NPT1) is a selective arginine transporter that is essential for parasite survival and virulence. We also demonstrate that a homologue of *Tg*NPT1 from the malaria parasite *Plasmodium berghei* (*Pb*NPT1), shown previously to be essential for the sexual gametocyte stage of the parasite, is a cationic amino acid transporter. This reveals a role for cationic amino acid scavenging in gametocyte biology. Our study demonstrates a critical role for amino acid transporters in the survival, virulence and life cycle progression of these parasites.

[1] Research School of Biology, Australian National University, Canberra, Australian Capital Territory 2601, Australia. [2] College of Public Health, Medical and Veterinary Sciences, James Cook University, Smithfield, Queensland 4878, Australia. [3] John Curtin School of Medical Research, Australian National University, Canberra, Australian Capital Territory 2601, Australia. [4] Queensland Tropical Health Alliance Research Laboratory, Australian Institute of Tropical Health and Medicine, James Cook University, Smithfield, Queensland 4878, Australia. Correspondence and requests for materials should be addressed to K.K. (email: kiaran.kirk@anu.edu.au) or to G.G.v.D. (email: giel.vandooren@anu.edu.au).

Apicomplexan parasites include the causative agents of malaria (Plasmodium spp) and toxoplasmosis (T. gondii). Ancestral apicomplexans were free-living marine algae that synthesized most of the organic molecules required for their survival[1]. As they evolved to become intracellular parasites, apicomplexans lost many biosynthetic pathways and became reliant on their hosts as a nutrient source[2]. Intracellular parasites take up nutrients such as sugars, nucleosides and nucleobases, vitamins and amino acids from their host cell, doing so via membrane transport proteins. Despite the role played by nutrient scavenging in sustaining the growth and development of apicomplexan parasites, very few of the transporter proteins involved have been characterized to date, and most of those that have been characterized are homologues of equivalent transporters in other organisms, such as mammals, yeast and plants[3–7]. The transporters responsible for amino acid uptake in apicomplexans are unknown, despite amino acids being essential nutrients in the disease-causing life stages of these organisms[8–10].

Bioinformatic surveys of apicomplexan genomes have identified numerous candidate transporters[11,12]. For some, homologies to transporters characterized in other organisms provide insight into their function. However there remain many, so-called 'orphan', transporters with unknown substrate affinities[11,13]. A recent, large-scale study of putative orphan transporters from Plasmodium berghei identified 19 such proteins that are essential at some stage of the parasite life cycle[13]. Among these were several 'novel putative transporters' (NPTs), members of an apicomplexan-specific family of putative transporter proteins, comprising five members in Plasmodium spp[11] and 16 in T. gondii (K. Parker, E.R., K.K. and G.v.D., manuscript in preparation). In P. berghei, three of the five NPTs (termed PbNPT1, PbMFR4 and PbMFR5) are essential for transmission through the mosquito stages of the life cycle[13]. PbNPT1 was shown previously to be a plasma membrane protein that is essential for the development of sexual-stage gametocytes[14]. Despite the apparent importance of the NPTs for parasite biology, the substrates, functions and physiological roles of these proteins remain unknown.

In this study, we demonstrate that an NPT protein from T. gondii, termed TgNPT1, functions as a highly selective arginine transporter, essential for parasite growth and virulence. We also show that T. gondii has a second broader-specificity cationic amino acid uptake pathway that transports both arginine and lysine, but which is not sufficient to support parasite growth under physiological conditions. PbNPT1, a malaria parasite homologue of TgNPT1, functions as a cationic amino acid transporter, implicating cationic amino acid scavenging in gametocyte biology. Our findings demonstrate that the previously uncharacterised, apicomplexan-specific NPT protein family includes amino acid transporters that are essential for survival of these parasites, and point to a critical role for the uptake of cationic amino acids in the survival and virulence of these parasites.

## Results

### TgNPT1 is important for *in vitro* growth of *T. gondii* parasites.
Previous studies identified PbNPT1 as being essential for gametocyte development in P. berghei[13,14]; however, its function remains unknown. Here, we explored the possibility that PbNPT1 transports nutrients or other substrates critical for gametocyte biology. In doing so, we turned first to PbNPT1 homologues in T. gondii, an experimentally tractable relative of Plasmodium parasites. Using the Basic Local Alignment Search Tool (BLAST) on www.toxodb.org we searched the T. gondii genome for homologues of PbNPT1. The top hit was to the gene TGME49_215490, referred to hereafter as TgNPT1. Like other NPTs, TgNPT1 is predicted to contain 12 transmembrane domains, and includes a signature sequence of the major facilitator superfamily of transporter proteins (Supplementary Fig. 1; ref. 15). To confirm expression of TgNPT1, we replaced the 3′ end of the TgNPT1 open reading frame with a hemagglutinin (HA) epitope tag in TATi/Δku80 strain parasites[16] and confirmed correct integration by polymerase chain reaction (PCR) analysis (Supplementary Fig. 2a,b). Western blotting revealed that TgNPT1-HA has a molecular mass of ∼40 kDa, below the calculated mass of 58 kDa (Fig. 1a). The reason for this discrepancy was not investigated further. It is possible that there is proteolytic processing of TgNPT1 at the N-terminus, or that the TgNPT1 protein migrates faster than predicted on SDS–polyacrylamide gel electrophoresis (SDS–PAGE), a common feature of hydrophobic membrane proteins[17]. We attempted to integrate a 5′ epitope tag into the TgNPT1 locus, but were unable to recover genetically modified parasites, precluding further analysis of N-terminal processing. An immunofluorescence assay revealed that TgNPT1-HA localizes to the periphery of the parasite, as well as to some internal structures (Fig. 1b). The peripheral localization overlapped with that of SAG1, a marker for the T. gondii plasma membrane, consistent with TgNPT1 being a plasma membrane protein.

To determine whether TgNPT1 is essential for T. gondii growth, we replaced the native promoter of TgNPT1 with an anhydrotetracycline (ATc)-regulated promoter (Supplementary Fig. 2c; ref. 16). This promoter enables inducible silencing of gene transcription on addition of ATc. The resultant parasite strain was termed iTgNPT1. Correct integration of the ATc-regulated promoter was confirmed by PCR analysis (Supplementary Fig. 2d). To determine the extent of TgNPT1 protein knockdown following addition of ATc, we introduced a 3′-HA tag into the iTgNPT1 locus and grew parasites for 0–24 h in the presence of ATc. A western blot of protein extracts revealed no detectable TgNPT1-HA protein after 24 h exposure of parasites to ATc (Fig. 1c; Supplementary Fig. 3). To measure growth of iTgNPT1 parasites, we introduced a Tomato Red Fluorescent Protein into the iTgNPT1 strain (generating the strain iTgNPT1/Tomato) and performed a fluorescence growth assay[18,19]. Parasites were grown in Dulbecco's Modified Eagle's medium (DMEM) for 9 days in the absence or presence of ATc. Parasites grew normally in the absence of ATc, but growth was severely impaired in the presence of ATc (Fig. 1d). Complementation of the iTgNPT1/Tomato parasite strain with a constitutively expressed copy of TgNPT1 (cTgNPT1) restored growth of the parasites in the presence of ATc (Fig. 1e), indicating that the growth phenotype observed in the mutant is solely the result of TgNPT1 knockdown. We conclude that TgNPT1 plays an essential role in the growth of parasites in DMEM.

We hypothesized that TgNPT1 is a transporter for a nutrient that is essential for parasite survival, and that incubation of TgNPT1-deficient T. gondii parasites in the presence of an increased concentration of that nutrient might overcome the growth defect observed. With this in mind, we prepared a 'homemade' culture medium, allowing us to modify the concentrations of candidate substrates. Roswell Park Memorial Institute 1640 (RPMI) medium is a commonly used growth medium, for which the individual components (for example, vitamins and amino acids) are commercially available. RPMI was therefore used as a base for the homemade medium. In preliminary experiments to determine whether T. gondii parasites could grow in RPMI, we cultured iTgNPT1/Tomato parasites for 7 days in standard RPMI in the absence and presence of

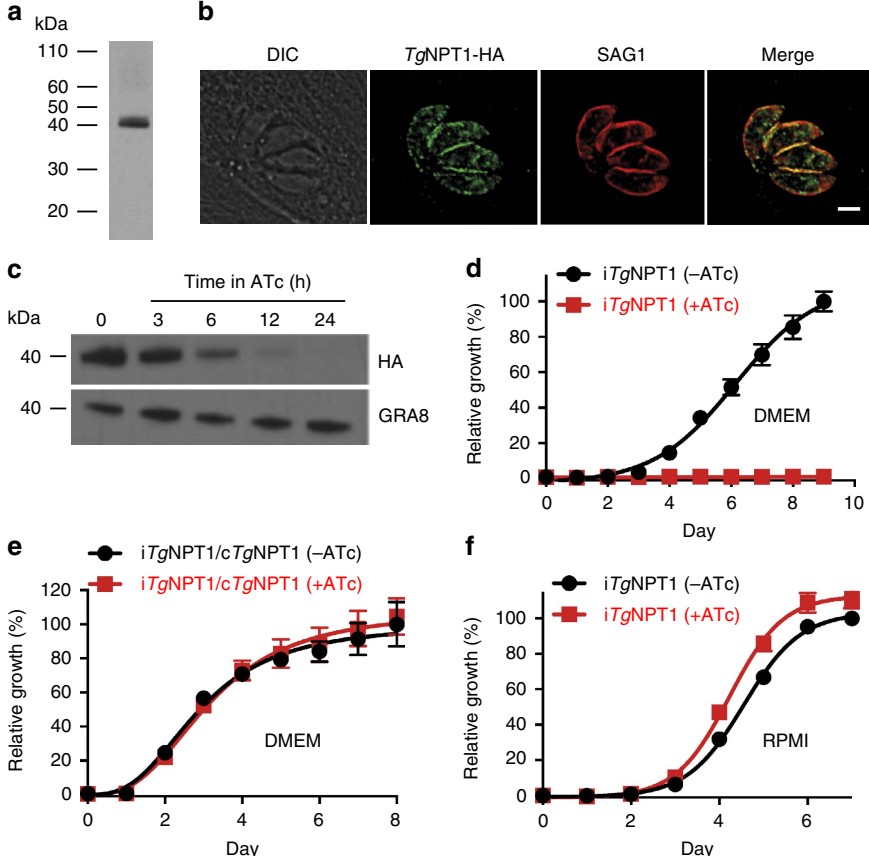

**Figure 1 | TgNPT1 is a plasma membrane protein essential for parasite growth in DMEM but not in RPMI.** (**a**) Western blot analysis of TgNPT1-HA, probed with anti-HA antibodies. (**b**) Immunofluorescence assay of TgNPT1-HA (green) reveals partial colocalization with the plasma membrane marker SAG1 (red) (Pearson's Correlation Coefficient mean ± SD = 0.81 ± 0.04, n = 6). The scale bar is 2 μm. (**c**) Western blot analysis demonstrating iTgNPT1-HA knockdown in the presence of ATc. Parasites were grown for 0, 3, 6, 12 and 24 h in the presence of ATc. GRA8 is a loading control. (**d–f**) Fluorescence growth assays for iTgNPT1 parasites (**d,f**) and iTgNPT1 parasites complemented with constitutively-expressed TgNPT1 (**e**; iTgNPT1/cTgNPT1), grown in DMEM (**d,e**) or RPMI (**f**), in the absence (black) or presence (red) of ATc. Growth is expressed as a percentage of that measured in parasites grown in the absence of ATc on the final day of the experiment. The data shown are averaged from three technical replicates (± s.d.), and are representative of those obtained in three biological replicates.

ATc. Unexpectedly, and in contrast to our findings with parasites grown in DMEM, we found that iTgNPT1/Tomato parasites grew normally in RPMI medium, both in the absence and presence of ATc (Fig. 1f). This serendipitous finding suggested that a difference in the composition of RPMI compared with DMEM modulated the growth of parasites subjected to TgNPT1 knockdown.

**Growth of parasites lacking TgNPT1 is modulated by arginine.** The differential growth effects seen in RPMI and DMEM were explored further in parasites in which the entire TgNPT1 gene was replaced with a selectable marker through double homologous recombination (Supplementary Fig. 2e). Disruption of the TgNPT1 locus was confirmed by PCR analysis (Supplementary Fig. 2f). The resultant 'knockout' strain (termed Δnpt1) grew normally in RPMI medium, but exhibited a severe growth defect in DMEM (Fig. 2a,b), consistent with our observations of the inducible knockdown line. By contrast, wild type (WT) parasites grew normally in both RPMI and DMEM (Fig. 2a,b).

To identify the component(s) of RPMI that enabled the growth of parasites deficient in TgNPT1, we prepared a range of 'hybrid' media that contained some components (vitamins and amino acids) at concentrations found in RPMI and some at concentrations found in DMEM. Media containing amino acids at the concentrations present in RPMI, but not media containing amino acids at the concentrations present in DMEM, supported the growth of iTgNPT1 parasites in the presence of ATc (Fig. 2c).

Arginine is present at a three-fold higher concentration in RPMI than in DMEM (Supplementary Fig. 4). To test whether arginine had a role in the differential growth phenotype, we grew WT and Δnpt1 parasites in RPMI containing arginine at the lower concentration present in DMEM (400 μM). In this medium the growth of Δnpt1 parasites, but not that of WT parasites, was impaired (Fig. 2d,e). We therefore investigated the dependence of the growth of both WT and Δnpt1 parasites on arginine concentration. WT parasites showed growth impairment when the arginine concentration was reduced below ~50 μM (Fig. 2f), consistent with previous data indicating that T. gondii is auxotrophic for arginine[9]. At arginine concentrations above ~50 μM there was no growth impairment. By contrast, Δnpt1 parasites exhibited severe growth impairment when arginine levels were reduced below 1.15 mM (Fig. 2f). Ablation of TgNPT1 therefore impairs parasite growth under conditions of restricted arginine availability.

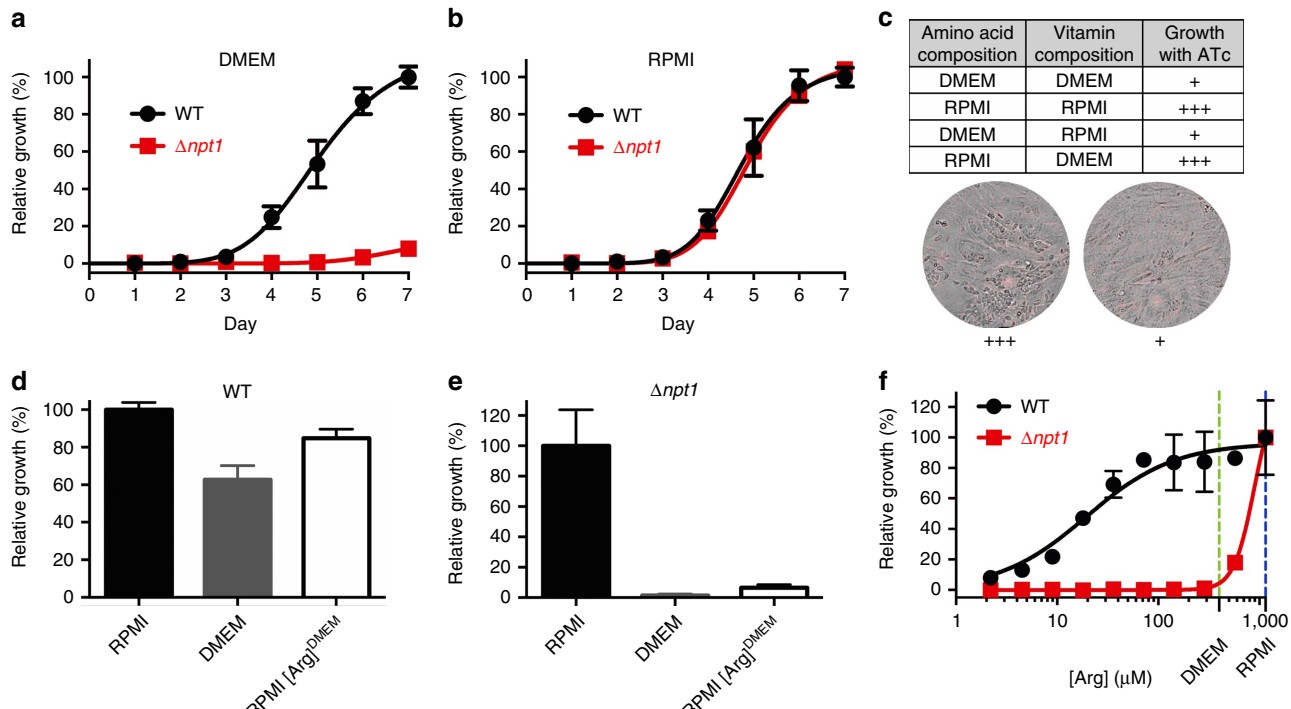

**Figure 2 | Growth of parasites lacking *Tg*NPT1 is modulated by arginine. (a,b)** Fluorescence growth assays for WT (black) and Δ*npt1* (red) parasites grown in DMEM (**a**) or RPMI (**b**). Growth is expressed relative to the maximum growth of WT parasites on the final day of the experiment under each of the conditions tested. The data shown are averaged from three technical replicates (±s.d.) and are representative of those obtained in three biological replicates. (**c**) Growth of i*Tg*NPT1 parasites in the presence of ATc in medium having the concentrations of amino acids and vitamins present in either RPMI or DMEM. The parasites grew well (+ + +) in medium containing the concentrations of amino acid present in RPMI, but poorly (+) in medium containing the concentrations of amino acids present in DMEM. (**d,e**) Growth of WT (**d**) and Δ*npt1* (**e**) parasites in the following media: RPMI (black), DMEM (grey) or RPMI containing the concentration of arginine present in DMEM (400 μM; RPMI[Arg]$^{DMEM}$; white). Parasites were cultured until those grown in RPMI reached mid-logarithmic stage. The growth of parasites in each medium is plotted as a percentage of the average growth of parasites in RPMI. The average of three technical replicates ± s.d. of a single experiment are shown. (**f**) Fluorescence growth assay for WT (black) and Δ*npt1* (red) parasites grown for 4 days in media containing a range of arginine concentrations. Parasite growth is expressed as a percentage of that measured at the highest arginine concentration (1.15 mM) for each parasite line. The arginine concentrations in DMEM and RPMI are indicated by the vertical green and blue dashed lines, respectively. The data shown are averaged from three technical replicates (±s.d.) and are representative of those obtained in three biological replicates.

***Tg*NPT1 is a selective arginine transporter**. Our data are consistent with *Tg*NPT1 functioning as an arginine transporter. To test this hypothesis, complementary RNA (cRNA) encoding *Tg*NPT1-HA was injected into *Xenopus laevis* oocytes, a well-validated heterologous expression system for the characterization of solute transporters[20]. *Tg*NPT1-HA protein localized to the plasma membrane of injected oocytes (Supplementary Fig. 5a). Measurements of the uptake of [14C]arginine ([14C]Arg) into *Tg*NPT1-HA cRNA-injected and uninjected oocytes over 30 min revealed that the uptake of arginine into oocytes expressing *Tg*NPT1-HA was eight-fold higher than that into control oocytes under the conditions tested (arginine concentration of 100 μM; Fig. 3a; Supplementary Fig. 5b), consistent with *Tg*NPT1 transporting arginine into the oocytes. We investigated the substrate specificity of *Tg*NPT1-HA by measuring the uptake of [14C]Arg in the presence of a range of unlabelled amino acids, each at a concentration of 1 mM, thereby testing the ability of each amino acid to compete with the uptake of [14C]Arg. The addition of 1 mM unlabelled arginine resulted in a significant, ∼85% inhibition of [14C]Arg uptake into oocytes expressing *Tg*NPT1-HA (Fig. 3b; $P < 0.0001$, ANOVA, $n = 3$), whereas there was no significant change for any of the other amino acids tested, including the cationic amino acids lysine and ornithine

(Fig. 3b; $P > 0.05$, ANOVA, $n = 3$). This pattern of inhibition is consistent with *Tg*NPT1 having a high degree of specificity for arginine. We next measured the rate of arginine uptake in *Tg*NPT1-HA-injected oocytes over a range of substrate concentrations (Fig. 3c). *Tg*NPT1-mediated uptake of arginine showed Michaelis-Menten kinetics, with an apparent $K_m$ of $88 \pm 15$ μM for arginine (mean ± s.e.m., $n = 3$) and a $V_{max}$ of $3.5 \pm 0.1$ pmol min$^{-1}$ oocyte$^{-1}$ (mean ± s.e.m., $n = 3$; Fig. 3b; Supplementary Fig. 5c).

**Characteristics of arginine transport via *Tg*NPT1**. To elucidate the mechanism of arginine transport by *Tg*NPT1, we measured the accumulation of [14C]Arg into *Tg*NPT1-expressing oocytes in media in which different ions were removed. Removal of Na$^+$, Cl$^-$, K$^+$, Mg$^{2+}$ or Ca$^{2+}$ had no effect on *Tg*NPT1-mediated [14C]Arg uptake measured over 30 min (Fig. 4a). The pH-sensitivity of *Tg*NPT1-mediated [14C]Arg transport was measured over a pH range of 5–9. [14C]Arg uptake into *Tg*NPT1-expressing oocytes was unaffected by pH over the pH range 5–8, but decreased significantly when the pH was increased above 8 (Fig. 4b).

The electrogenicity of arginine transport by *Tg*NPT1 was investigated using the electrophysiological two-electrode

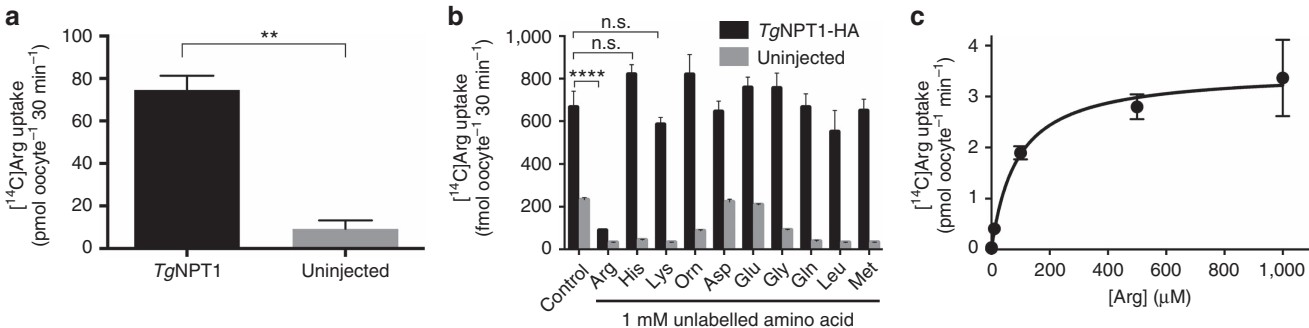

**Figure 3 | TgNPT1 is an arginine transporter.** (**a**) [14C]Arg uptake into *X. laevis* oocytes expressing TgNPT1-HA (black), or into uninjected controls (grey). The data are derived from the 30 min time point of Supplementary Fig. 5b. Uptake was measured in the presence of 100 μM unlabelled arginine and 289 nM [14C]Arg. The data are averaged from three independent experiments and are shown ± s.e.m. (\*\*$P<0.01$; Student's *t* test). (**b**) [14C]Arg uptake, measured over 30 min, into *X. laevis* oocytes expressing TgNPT1-HA (black) or into uninjected oocytes (grey), in the absence (control) or presence of a 1 mM concentration of a range of unlabelled amino acids, and 289 nM [14C]Arg. The data are averaged from three experiments, each conducted on oocytes from a different frog, and are shown ± s.e.m. (\*\*\*\*$P<0.0001$; n.s. = not significant; ANOVA. Where *P* is not indicated for comparisons between uptake in the presence and absence of unlabelled amino acids in TgNPT1-HA expressing oocytes, the differences are not significant). (**c**) Concentration-dependence of TgNPT1-mediated arginine transport in *X. laevis* oocytes expressing TgNPT1-HA. [14C]Arg uptake was measured over 30 min in the presence of varying concentrations of arginine (0–1 mM). At each of the arginine concentrations tested, uptake measured in uninjected oocytes was subtracted from that in oocytes expressing TgNPT1-HA to yield TgNPT1-mediated transport. The data were averaged from three experiments, each conducted on oocytes from a different frog, and are shown ± s.e.m. The Michaelis–Menten equation was fitted to the data by non-linear regression ($K_m = 88 \pm 16$ μM, mean ± s.e.m.; $V_{max} = 3.5 \pm 0.1$ pmol oocyte$^{-1}$ min$^{-1}$, mean ± s.e.m.).

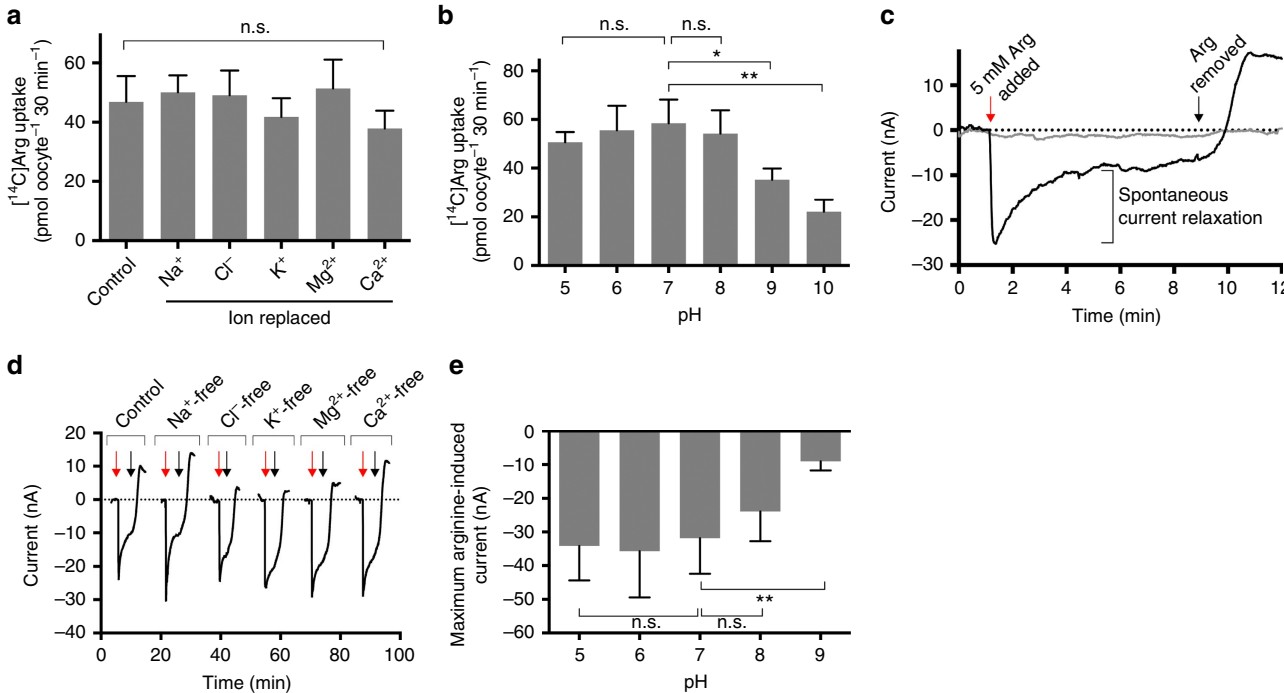

**Figure 4 | Characteristics of arginine transport by TgNPT1.** (**a,b**) Ion-dependence (**a**) and pH-dependence (**b**) of TgNPT1-mediated arginine transport in oocytes expressing TgNPT1-HA. The uptake of 0.4 μCi ml$^{-1}$ (1.1 μM) [14C]Arg was measured in the presence of 100 μM unlabelled arginine in oocytes, in either complete ND96 uptake buffer (control), media in which Na$^+$, Cl$^-$, K$^+$, Mg$^{2+}$, or Ca$^{2+}$ ions were replaced (**a**), or media pH-adjusted to pH 5–10 (**b**). Uptake measured in uninjected control oocytes was subtracted from that measured in TgNPT1-HA-expressing oocytes to yield the TgNPT1-mediated uptake. All data were averaged from three independent experiments, each conducted on oocytes from a different frog (\*$P<0.05$, \*\*$P<0.01$; n.s. = not significant; ANOVA). (**c**) A representative trace for the arginine-induced current in a TgNPT1-HA-expressing oocyte (black) and an uninjected oocyte (grey). The red arrow indicates the point of addition of 5 mM arginine to the ND96 medium and the black arrow indicates the point at which arginine was removed. The spontaneous current relaxation following the addition of 5 mM arginine is indicated. (**d**) Representative traces of arginine-induced currents in TgNPT1-HA-expressing oocytes, suspended in complete ND96 medium (control) and in media lacking Na$^+$, Cl$^-$, K$^+$, Mg$^{2+}$ or Ca$^{2+}$ ions. For all current traces, the resting current before arginine addition was set to 0. Red arrows indicate the points of addition of 5 mM arginine to the ND96 medium and black arrows indicate the points at which arginine was removed. The traces are representative of $n=8$ oocytes under each of the conditions tested. (**e**) pH dependence of the maximum arginine-induced current in oocytes expressing TgNPT1-HA. Oocytes were equilibrated in media of the relevant pH and the maximum amplitude of the current following the addition of 5 mM arginine was measured. For each pH, the mean arginine-induced current is shown ± s.d. ($n=8$; \*\*$P<0.01$; n.s. = not significant; ANOVA).

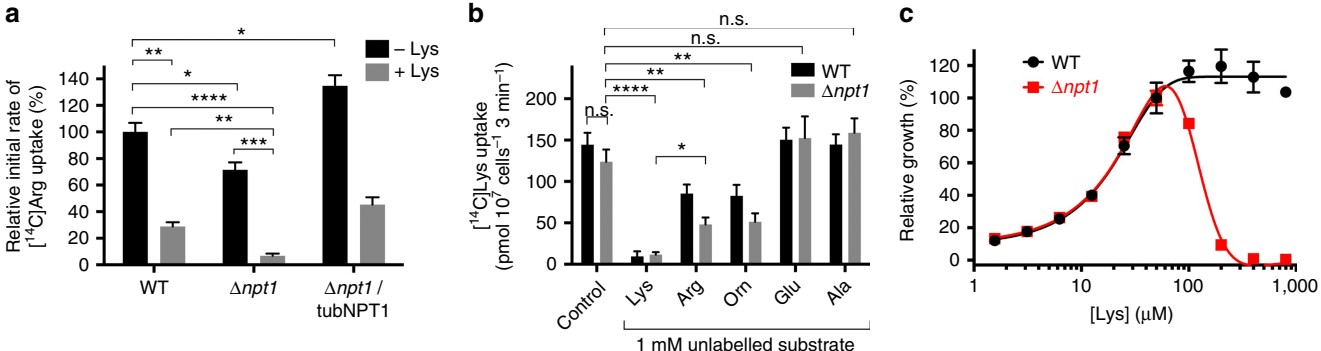

**Figure 5 | Arginine uptake into *T. gondii* is mediated by *Tg*NPT1 and by a *Tg*NPT1-independent cationic amino acid uptake pathway. (a)** [$^{14}$C]Arg
uptake in WT, Δ*npt1* and Δ*npt1*/tubNPT1 parasites in the absence (black) and presence (grey) of 80 μM unlabelled lysine, expressed as a percentage of the
initial rate of [$^{14}$C]Arg uptake in WT parasite measured in the absence of lysine. Uptake was measured in parasites suspended in PBS containing 10 mM
glucose, 40 μM unlabelled arginine and 0.1 μCi ml$^{-1}$ (289 nM) [$^{14}$C]Arg. The initial rates of [$^{14}$C]Arg uptake were derived from the initial slopes of the
time courses shown in Supplementary Fig. 7a. The mean initial rate of [$^{14}$C]Arg uptake in WT parasite measured in the absence of lysine was
900 ± 61 pmol 10$^7$ cells$^{-1}$ min$^{-1}$ (mean ± s.e.m.; $n = 3$). The data shown represent the mean ± s.e.m. from three independent experiments
(*$P < 0.05$; **$P < 0.01$; ***$P < 0.001$; ****$P < 0.0001$; Student's $t$ test). **(b)** [$^{14}$C]Lys uptake in *T. gondii*. The uptake of 0.1 μCi ml$^{-1}$ (307 nM) [$^{14}$C]Lys
(in the presence of 50 μM unlabelled lysine) was measured over 3 min (within the initial linear phase of uptake; Supplementary Fig. 7b) in WT (black) and
Δ*npt1* (grey) parasites, suspended either in the presence or absence of a 1 mM concentration of the cationic amino acids lysine (Lys), arginine (Arg) or
ornithine (Orn), the anionic amino acid glutamate (Glu), or the small neutral amino acid alanine (Ala). The results are averaged from those obtained in
three separate experiments ± s.e.m. (*$P < 0.05$; **$P < 0.01$; ****$P < 0.0001$; n.s. = not significant; ANOVA). **(c)** Fluorescence growth assay for WT (black)
and Δ*npt1* (red) parasites cultured for 4 days in media having a range of lysine concentrations and a constant 400 μM arginine. Growth is expressed as a
percentage of that measured at 50 μM lysine for each parasite strain. The data shown are averaged from three technical replicates (shown ± s.d.) and are
representative of those obtained in three biological replicates.

voltage-clamp technique[21]. Addition of 5 mM arginine
to oocytes expressing *Tg*NPT1 resulted in an inward current
that reached a maximum of 22 ± 5 nA (mean ± s.e.m., $n = 20$;
Fig. 4c, black trace). The maximum peak current was followed
by a spontaneous relaxation (Fig. 4c), reminiscent of
the behaviour of the arginine-induced current observed in
oocytes expressing the human cationic amino acid transporter
hCAT-2a (ref. 22). Addition of arginine to uninjected oocytes
did not induce a current (Fig. 4c, grey trace). The data are
consistent with *Tg*NPT1 mediating the electrogenic uptake of
arginine.

The dependence of the arginine-induced current on the ionic
composition of the medium was tested by measuring the current
in oocytes exposed to media from which Na$^+$, Cl$^-$, K$^+$, Mg$^{2+}$
or Ca$^{2+}$ was absent. In each of the media tested, the arginine-
induced current was similar (Fig. 4d), consistent with electrogenic
*Tg*NPT1-mediated arginine uptake occurring via an ion-inde-
pendent mechanism. The dependence of the arginine-induced
current on pH was tested by measuring the current in media
of varying pH. The amplitude of the arginine-induced
inward current was largely insensitive to pH in the pH range
5–8 ($P > 0.05$, ANOVA, $n = 8$), but was significantly decreased
at pH 9 ($P < 0.01$, ANOVA, $n = 8$; Fig. 4e; Supplementary
Fig. 6a,b). This mirrored the effects of pH on [$^{14}$C]Arg uptake
(Fig. 4b).

Altogether, these data demonstrate that *Tg*NPT1 mediates
the electrogenic transport of arginine via a mechanism
that is independent of Na$^+$, Cl$^-$, K$^+$, Mg$^{2+}$ and Ca$^{2+}$,
and which is sensitive to pH at pH values >8. Whether
the electrogenicity arises solely from the transport of the
cationic amino acid, or whether it might involve the transport
of H$^+$ is unclear. The observed maximum transport rate derived
from the uptake of [$^{14}$C]Arg equates to a current of 6 nA
(1 nA = 36 pmol charges h$^{-1}$); this is lower than the currents
measured under voltage clamp conditions, and might be
explained by a contribution of H$^+$ to the observed currents.

**T. gondii has a TgNPT1-independent arginine uptake pathway.**
To measure the contribution of *Tg*NPT1 to arginine uptake
in *T. gondii*, we compared the uptake of [$^{14}$C]Arg into Δ*npt1*
parasites with that into WT parasites (Fig. 5a; Supplementary
Fig. 7a). The initial rate of [$^{14}$C]Arg uptake in Δ*npt1* parasites was
reduced to 71 ± 5% (mean ± s.e.m., $P < 0.05$, Student's $t$ test,
$n = 3$) of that in the WT strain, consistent with *Tg*NPT1
contributing significantly to the uptake of arginine into *T. gondii*
parasites under the conditions tested. In Δ*npt1* parasites
complemented with an ectopic copy of *Tg*NPT1, expressed
from the constitutive α-tubulin promoter, the rate of [$^{14}$C]Arg
uptake was higher than that in WT parasites (Δ*npt1*/tubNPT1;
Fig. 5a, $P < 0.05$, Student's $t$ test, $n = 3$).

These data suggest the presence in the parasite of one or
more *Tg*NPT1-independent arginine uptake pathways that
account for the remaining arginine uptake detected in parasites
lacking *Tg*NPT1. To probe the substrate selectivity of the
*Tg*NPT1-independent arginine uptake pathway(s), we measured
[$^{14}$C]Arg uptake into WT, Δ*npt1* and Δ*npt1*/tubNPT1 parasites
in the presence of unlabelled lysine (80 μM). Unlabelled
lysine reduced the initial rate of uptake of [$^{14}$C]Arg into
WT parasites to 29 ± 3% (mean ± s.e.m., $P < 0.01$, Student's
$t$ test, $n = 3$) of that measured in the absence of lysine, and
reduced the rate of [$^{14}$C]Arg uptake into Δ*npt1* parasites
to 7 ± 2% (mean ± s.e.m., $P < 0.0001$, Student's $t$ test, $n = 3$)
of that in WT parasites in the absence of lysine (Fig. 5a).
This is consistent with unlabelled lysine interacting with
the *Tg*NPT1-independent arginine uptake pathway(s), and
thereby competing with [$^{14}$C]Arg for uptake into parasites.

The ability of the alternative arginine uptake pathway(s) to
transport lysine was tested directly by measuring the initial rate
of uptake of [$^{14}$C]lysine ([$^{14}$C]Lys) in WT and Δ*npt1* parasites
(Fig. 5b; Supplementary Fig. 7b). There was no significant
difference between the rate of [$^{14}$C]Lys uptake in WT and
Δ*npt1* parasites (Fig. 5b, control; $P > 0.05$, ANOVA, $n = 3$).
This suggests that *Tg*NPT1 does not contribute to lysine transport

in the parasite, and is consistent with the observation that lysine does not compete with the uptake of [$^{14}$C]Arg into oocytes expressing $Tg$NPT1 (Fig. 3a). [$^{14}$C]Lys uptake by both WT and $\Delta npt1$ parasites was inhibited by the addition of 1 mM of the unlabelled cationic amino acids lysine, arginine and ornithine (Fig. 5b; $P < 0.01$, ANOVA, $n = 3$), consistent with the $Tg$NPT1-independent arginine uptake pathway(s) being able to transport a range of cationic amino acids, including arginine, lysine and ornithine. By contrast, the addition of 1 mM of the anionic amino acid glutamate or the neutral amino acid alanine had no effect on [$^{14}$C]Lys uptake in either WT or $\Delta npt1$ parasites (Fig. 5b; $P > 0.05$, ANOVA, $n = 3$), consistent with neither of these amino acids competing for the transporter.

The relationship between arginine and lysine uptake and parasite growth was investigated by measuring the growth of WT and $\Delta npt1$ parasites over a range of lysine concentrations. Parasites were grown in RPMI containing 1.6–800 μM lysine, together with 400 μM arginine (constant throughout). For WT parasites, growth was maximal at lysine concentrations > 50 μM but decreased progressively as the lysine concentration was reduced below 50 μM (Fig. 5c). At a lysine concentration of 50 μM, the growth of $\Delta npt1$ parasites was comparable to that of WT parasites and, as for WT parasites, growth decreased progressively as the lysine concentration was reduced below 50 μM. However, in marked contrast to WT parasites, the growth of $\Delta npt1$ parasites decreased as the lysine concentration was increased above 50 μM, with negligible growth observed at lysine concentrations ≥ 400 μM (Fig. 5c).

Mammalian cells are auxotrophic for lysine, and it is conceivable that the impairment of parasite growth in media with lysine concentrations below 50 μM is the result of reduced host mammalian cell viability. The alternative is that the parasite itself is auxotrophic for lysine. The *T. gondii* genome harbours at least some genes that encode for enzymes involved in lysine synthesis[23,24], although the functionality of this pathway has not been demonstrated. The identity of the general cationic amino acid transporter(s) responsible for the uptake of lysine remains an open question. There are 15 other NPT-family transporters in the *T. gondii* genome, and one or more of these may function in this role. Identifying and characterizing the mechanism(s) of lysine uptake will reveal the importance of lysine scavenging in *T. gondii* growth.

The growth inhibition of the $\Delta npt1$ mutant (Fig. 5c) at high-lysine concentrations (> 50 μM) may be accounted for in terms of the model represented in Fig. 6, in which the parasite has (at least) two arginine uptake systems, differing with respect to their relative affinities for arginine and lysine. $Tg$NPT1 is an arginine-specific transporter that mediates arginine uptake regardless of the levels of other cationic amino acids. In addition, parasites harbour one or more general cationic amino acid transport pathways that transport both arginine and lysine. If the arginine concentration in the medium is high relative to that of lysine (as is the case in RPMI, in which the [Arg]:[Lys] ratio is 5.25), there is sufficient uptake of arginine through the general transport pathway to enable parasite growth (Fig. 6). However, when the balance of [Arg]:[Lys] is shifted in the other direction (as is the case in DMEM, in which the [Arg]:[Lys] ratio is 0.5, a 10-fold reduction relative to RPMI), lysine competes with arginine for uptake through the general pathway, and parasites become reliant on $Tg$NPT1 to take up sufficient arginine to support growth (Fig. 6). In $\Delta npt1$ parasites, the $Tg$NPT1-independent pathway(s) can transport sufficient arginine to maintain parasite growth only under conditions in which the ratio of [Arg]:[Lys] is high enough that competition by lysine does not restrict the

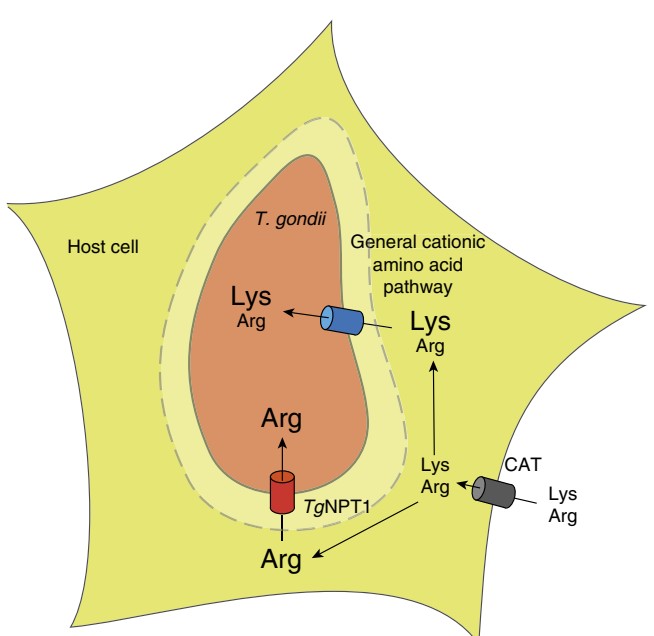

**Figure 6 | A model for arginine transport into *T. gondii* parasites.** Cationic amino acids such as arginine and lysine enter host cells through cationic amino acid transporters (CAT; grey cylinder). Arginine in the host cell cytosol crosses the parasitophorous vacuole membrane (dashed line) through non-selective pores[57], and is taken up by the parasite through two pathways. $Tg$NPT1 (red cylinder) is a selective arginine transporter, and serves as a major route for arginine uptake *in vivo*. A general cationic amino acid transport system (blue cylinder) facilitates the $Tg$NPT1-independent uptake of both arginine and lysine.

uptake of arginine through the general pathway to below the level required for normal growth.

The model assumes that the ratio of arginine to lysine in the host cell cytosol is similar to that in the extracellular medium. Cationic amino acids are taken up into mammalian cells by a number of broad-specificity cationic amino acid transporters (CATs; Fig. 6; refs 25,26). A previous study demonstrated a strong correlation between extra- and intra-cellular arginine concentrations in mammalian cells[27]. An older study noted that the ratio of [Arg]:[Lys] in human plasma is 0.44, and that the ratio in muscle tissue is 0.44 (ref. 28), suggesting a close similarity between the [Arg]:[Lys] ratio in the extracellular environment and that in the cell cytosol. The [Arg]:[Lys] ratio imposed in the extracellular milieu is therefore likely to be reflected in the intracellular environments to which *T. gondii* parasites are exposed. Regardless, our data indicate that the relative concentrations of lysine and arginine in the medium modulate the growth of $\Delta npt1$ parasites, and highlight the limitations of *in vitro* culture conditions for elucidating the importance of proteins such as $Tg$NPT1 for parasite survival.

**$Tg$NPT1 is essential for parasite virulence.** To assess whether $Tg$NPT1 is essential for parasite virulence *in vivo*, we infected BALB/c mice intraperitoneally with 10$^6$ WT, $\Delta npt1$ or $\Delta npt1$/tubNPT1 parasites and monitored disease progression. All mice infected with WT or $\Delta npt1$/tubNPT1 parasites exhibited symptoms of toxoplasmosis and were euthanised within 8 days post-infection (Fig. 7). In contrast, mice infected with $\Delta npt1$ parasites exhibited no signs of toxoplasmosis over the course of the entire 57 day experiment (Fig. 7). $Tg$NPT1 is therefore essential for parasite virulence under *in vivo* conditions.

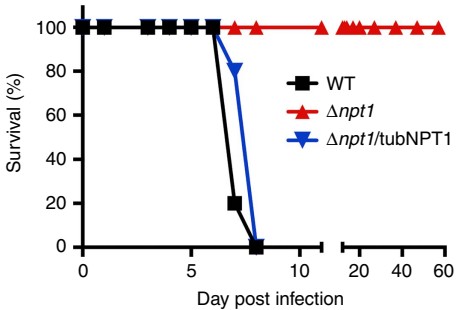

**Figure 7 | *Tg*NPT1 is essential for parasite virulence *in vivo*.** Five Balb/c mice were infected with $10^6$ WT (black), Δ*npt1* (red) or Δ*npt1*/tubNPT1 (blue) parasites and monitored for symptoms of toxoplasmosis. The data for WT and Δ*npt1* parasite infections are derived from two independent experiments, whereas data for Δ*npt1*/tubNPT1 infections are derived from a single experiment.

The finding that *Tg*NPT1 is important for parasite virulence might be understood in terms of the ratio of [Arg]:[Lys] in mouse plasma, estimated to be ∼0.1 (ref. 29). Under these conditions it is predicted (on the basis of the *in vitro* data; Fig. 5a,c) that competition by lysine will limit the uptake of arginine through the general, *Tg*NPT1-independent, cationic amino acid uptake pathway(s) to a level below that required to support growth. The *in vivo* data are therefore consistent with the model for cationic amino acid uptake in *T. gondii* represented in Fig. 6, and with *Tg*NPT1 providing an essential route for the scavenging of arginine from infected hosts.

Arginine uptake in mammalian cells is mediated by broad-specificity cationic amino acid transporters[26]. By contrast, arginine uptake in *Leishmania* parasites, which are intracellular for a portion of their life cycle, is mediated by a highly selective arginine transporter[30], much like *Tg*NPT1-mediated arginine uptake in *T. gondii*. Intracellular parasites are in competition with their host cells for available arginine, perhaps necessitating their having a selective, high-affinity arginine transporter to scavenge sufficient amounts of this amino acid for parasite survival.

***Pb*NPT1 is a cationic amino acid transporter**. To investigate whether the *P. berghei* homologue of *Tg*NPT1 (*Pb*NPT1) can also transport arginine, we expressed HA-tagged *Pb*NPT1 in *Xenopus* oocytes (Supplementary Fig. 5d), and measured the uptake of [$^{14}$C]Arg over 30 min. Under the conditions of the experiment, oocytes expressing *Pb*NPT1-HA showed a significant, ∼4-fold increase in the uptake of [$^{14}$C]Arg, relative to uninjected control oocytes (Fig. 8a, $P < 0.05$, Student's $t$ test, $n = 3$; Supplementary Fig. 5e), consistent with *Pb*NPT1 functioning as an arginine transporter. The substrate specificity of *Pb*NPT1 was investigated by measuring the uptake of [$^{14}$C]Arg in the presence of a 1 mM concentration of a range of unlabelled amino acids. The uptake of [$^{14}$C]Arg was reduced by the addition of unlabelled arginine or lysine ($P < 0.0001$, ANOVA, $n = 3$) and, to a lesser extent, by unlabelled ornithine ($P < 0.001$, ANOVA, $n = 3$) or histidine ($P < 0.05$, ANOVA, $n = 3$, Fig. 8b), consistent with *Pb*NPT1 interacting with a range of cationic amino acids and therefore having a broader substrate specificity than *Tg*NPT1. Uptake of [$^{14}$C]Lys into oocytes expressing *Pb*NPT1 was increased 4-fold relative to that into uninjected control oocytes (Fig. 8a, $P < 0.05$, Student's $t$ test, $n = 3$; Supplementary Fig. 5f), consistent with *Pb*NPT1 also

functioning as a lysine transporter. For both arginine and lysine, the concentration-dependence of *Pb*NPT1-induced uptake showed Michaelis–Menten kinetics (Fig. 8c,d; Supplementary Fig. 5g,h), with the transporter having a higher affinity for arginine (apparent $K_m = 41 \pm 9\,\mu M$, mean ± s.e.m., $n = 3$) than for lysine (apparent $K_m = 130 \pm 26\,\mu M$, mean ± s.e.m., $n = 3$).

To investigate the role of *Pb*NPT1 in the uptake of cationic amino acids into *P. berghei* parasites, we generated a *Pb*NPT1 knockout parasite strain (termed Δ*Pbnpt1*; Supplementary Fig. 2g,h) as described previously[14]. In the previous study it was reported that Δ*Pbnpt1* parasites grew normally in asexual blood stages of the parasite, but were defective in gametocyte development and progression into the insect stages of the *Plasmodium* life cycle[14]. Consistent with this, we found that Δ*Pbnpt1* parasites were significantly impaired in the formation of microgametes, the motile stage of the sexual cycle (Fig. 9a, $P < 0.05$, Student's $t$ test, $n = 3$). The uptake of both [$^{14}$C]Arg and [$^{14}$C]Lys into asexual-stage Δ*Pbnpt1* parasites (isolated from their host erythrocytes by saponin permeabilisation of the host cell and parasitophorous vacuole membranes) was significantly decreased, to $12 \pm 3\%$ and $13 \pm 1\%$ (mean ± s.e.m.; $P < 0.05$, ANOVA, $n = 3$) respectively, of uptake into WT parasites (Fig. 9b). The uptake and incorporation of the glucose analogue [1-$^{14}$C]2-deoxy-glucose (2-DOG) was similar in WT and Δ*Pbnpt1* parasites (Fig. 9b, $P > 0.05$, ANOVA, $n = 3$). This indicates that Δ*Pbnpt1* parasites remained metabolically active, and suggests that defects in the uptake of cationic amino acids was not the result of general defects in parasite viability. Altogether, these findings are consistent with *Pb*NPT1 playing a major and specific role in the uptake of cationic amino acids into the intracellular malaria parasite.

Intraerythrocytic asexual stages of the human malaria parasite *P. falciparum* have been shown to take up exogenous arginine via an unidentified pathway that is inhibited by other cationic amino acids[31,32]. The functional characteristics of *Pb*NPT1 expressed in *Xenopus* oocytes are very similar to those of the pathway mediating the uptake of arginine into *P. falciparum* parasites[31], consistent with the *P. falciparum* orthologue of *Pb*NPT1 (PF3D7_0104800) serving as the major arginine transporter in asexual-stage *P. falciparum* parasites. *P. falciparum* infection causes depletion of arginine in the host erythrocyte[31], and *Plasmodium* infection leads to hypoargininaemia, a depletion of arginine in the plasma of the host that contributes to poor disease outcomes such as cerebral malaria[33,34]. *Plasmodium* NPT1 conceivably plays a role in *Plasmodium*-induced hypoargininaemia.

Our data point to a role for the uptake of one or more cationic amino acids in gametocyte development in *P. berghei*. With an abundance of amino acids available through haemoglobin digestion during the early sexual stages of the life cycle[10,35], the requirement for a cationic amino acid transporter for gametocyte development is unexpected. A recent study found that depletion of the amino acid asparagine in blood plasma leads to defects in the development of the sexual stages of *P. berghei*[36]. This mirrors our findings with cationic amino acids, and suggests that gametocytes may have a greater nutritional requirement for amino acids than can be met by host cell haemoglobin degradation alone. Alternatively, arginine consumption by the parasite could limit the availability of arginine for host processes such as the synthesis of nitric oxide, an anti-parasitic host defence mechanism that is effective at targeting the gametocyte stage of the parasite[37]. These hypotheses are the subject of ongoing investigations.

*T. gondii* and *P. falciparum* are auxotrophic for several amino acids at various stages of their life cycles[8–10,38]. Despite the importance of amino acid uptake, no amino acid

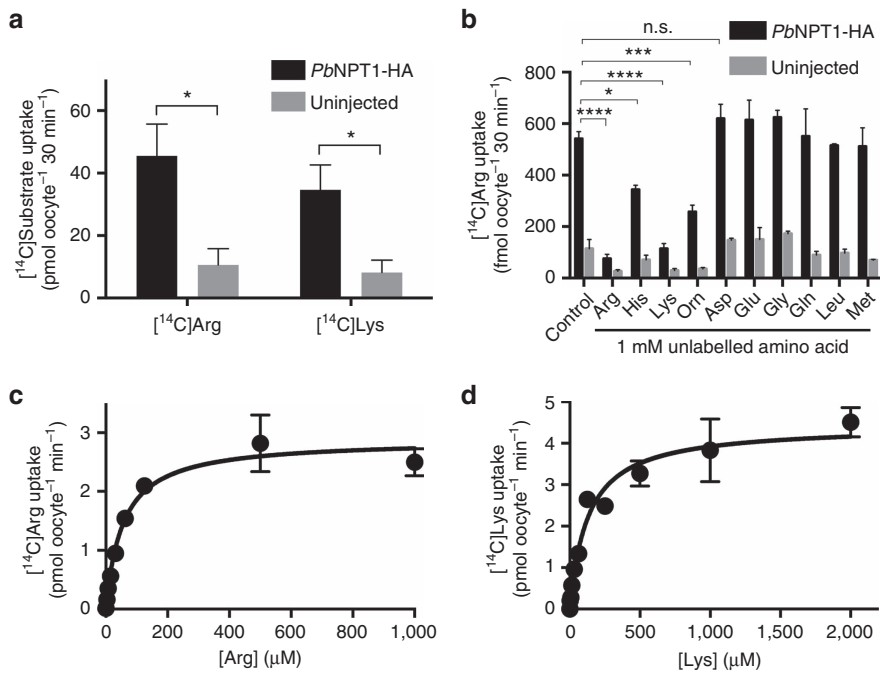

**Figure 8 | PbNPT1 is a cationic amino acid transporter. (a)** Uptake of [14C]Arg and [14C]Lys into *X. laevis* oocytes expressing PbNPT1-HA (black), or into uninjected controls (grey). The data are derived from the 30 min time points of Supplementary Fig. 5e,f. Uptake was measured in the presence of 289 nM [14C]Arg, 100 μM unlabelled arginine and 125 μM unlabelled methionine or 307 nM [14C]Lys, 100 μM unlabelled lysine and 500 μM unlabelled methionine. The data are averaged from three independent experiments and are shown ± s.e.m. (\*$P<0.05$; Student's *t* test). **(b)** Uptake of [14C]Arg into oocytes expressing PbNPT1-HA (black) or uninjected oocytes (grey), incubated in the presence of 289 nM [14C]Arg and in either the absence (control) or presence of a 1 mM concentration of a range of unlabelled amino acids, and 125 μM unlabelled methionine. The data are averaged from three experiments, each conducted on oocytes from a different frog, and are shown ± s.e.m. (\*$P<0.05$; \*\*\*$P<0.001$; \*\*\*\*$P<0.0001$; n.s. = not significant; ANOVA. Where *P* is not indicated for comparisons between uptake in the presence and absence of unlabelled amino acids in PbNPT1-HA expressing oocytes, the differences are not significant). **(c,d)** Concentration-dependence of PbNPT1-mediated arginine and lysine transport in *X. laevis* oocytes expressing PbNPT1-HA. The uptake of [14C]Arg **(c)** and [14C]Lys **(d)** was measured over 30 min, in the presence of 125 μM **(c)** or 500 μM **(d)** unlabelled methionine. At each of the concentrations tested, uptake measured in control (uninjected) oocytes was subtracted from that in oocytes expressing PbNPT1 to yield PbNPT1-mediated transport. The data were averaged from three experiments, each conducted on oocytes from a different frog, and are shown ± s.e.m. The Michaelis–Menten equation was fitted to the data by non-linear regression. For arginine, $K_m = 41 \pm 9$ μM, mean ± s.e.m.; $V_{max} = 2.6 \pm 0.1$ pmol oocyte$^{-1}$ min$^{-1}$, mean ± s.e.m. For lysine, $K_m = 130 \pm 26$ μM, mean ± s.e.m.; $V_{max} = 4.4 \pm 0.2$ pmol oocyte$^{-1}$ min$^{-1}$, mean ± s.e.m.

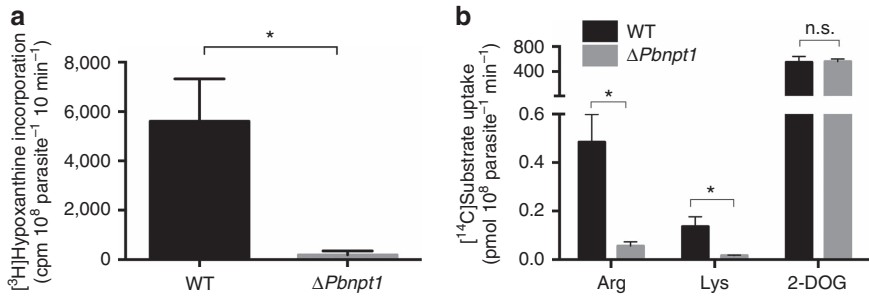

**Figure 9 | PbNPT1 is essential for microgametogenesis and mediates the uptake of cationic amino acids into *P. berghei* parasites. (a)** Quantification of microgamete formation in WT and *ΔPbnpt1* parasites by measuring [3H]hypoxanthine incorporation following induction of microgametogenesis by temperature shift and the addition of xanthurenic acid[54]. Data were averaged from three biological replicates, and are shown ± s.e.m. (\*$P<0.05$; Student's *t* test). **(b)** Uptake of [14C]Arg, [14C]Lys and [14C]2-DOG by asexual-stage WT (black) or *ΔPbnpt1* (grey) parasites, isolated from their host erythrocytes by saponin permeabilisation of the host erythrocyte and parasitophorous vacuole membranes. The isolated parasites were suspended in PBS containing either 0.1 μCi ml$^{-1}$ (289 nM) [14C]Arg, 0.1 μCi ml$^{-1}$ (307 nM) [14C]Lys, or 0.1 μCi ml$^{-1}$ (1.8 μM) [14C]2-DOG. Incorporation was measured after 4 min, a time point that is approximately within the linear uptake phase of arginine in *P. falciparum*[31]. Results are expressed as mean ± s.e.m., $n = 3$ (\*$P<0.05$; n.s. = not significant; ANOVA).

transporters have been characterized previously in apicomplexans. Arginine/cationic amino acid transporters from eukaryotes belong to the SLC7 or amino acid/auxin permease (AAAP) families of transporters (for example, refs 25,30).

These are fundamentally different, at both primary sequence and structural levels, to the major facilitator superfamily class to which *Tg*NPT1 and PbNPT1 belong[39]. This suggests that apicomplexans have evolved cationic amino acid transporters

independently of other major eukaryotic lineages. Apicomplexans lack SLC7 homologues, but harbour homologues of AAAP family transporters[12], although the functions of these remain to be elucidated. Orphan transporters, such as the NPTs, lack homologues in humans[11,13]. We and others have demonstrated that NPT-family proteins are essential for virulence and life cycle progression. These and other essential orphan transporters are therefore attractive targets for therapeutic interventions against apicomplexans.

## Methods

**Parasite strains and cultivation.** The TATi/Δku80 strain of *T. gondii*[16] was used as the 'wild type' parental strain for all lines generated in this study. *T. gondii* tachyzoites were cultured in human foreskin fibroblasts in Dulbecco's Modified Eagle's medium (DMEM) supplemented with 1% (v/v) fetal bovine serum (FBS) and antibiotics. *T. gondii* Δ*npt1* parasites, and derivatives thereof, were standardly cultured in Roswell Park Memorial Institute 1640 (RPMI) medium, supplemented with 1% (v/v) FBS and antibiotics. 'Homemade' media was prepared with the following salts: 0.265 g l$^{-1}$ CaCl$_2$.2H$_2$O, 0.20 g l$^{-1}$ MgSO$_4$.7H$_2$O, 0.40 g l$^{-1}$ KCl, 1.48 g l$^{-1}$ NaHCO$_3$, 6.40 g l$^{-1}$ NaCl and 0.234 g l$^{-1}$ NaH$_2$PO$_4$.2H$_2$O, in addition to commercially available RPMI vitamins (Sigma-Aldrich), 25 mM glucose, 1% (v/v) FBS, antibiotics, and concentrations of amino acids as specified in the text. Where appropriate, anhydrotetracycline (ATc) was added to a final concentration of 0.5 μg ml$^{-1}$.

For the infection of mice with *T. gondii*, WT, Δ*npt1*, and Δ*npt1*/tubNPT1 parasites were filtered through a 3 μm filter, washed once in sterile phosphate-buffered saline (PBS) and diluted to 10$^7$ cells ml$^{-1}$. Groups of five 6–8 week old, female BALB/c mice per condition (blindly and randomly assigned by a technician not familiar with details of the experiment) were injected intraperitoneally with 100 μl of parasites (10$^6$ cells) of each strain using a 26 or 27 gauge needle. Infections with WT and *npt1* parasites were performed in duplicate (10 mice in total for each condition), and infections with Δ*npt1*/tubNPT1 parasites were performed once (5 mice in total). Weight and mortality of the animals were recorded almost daily for a period of 57 days. Symptoms of toxoplasmosis including weight loss, lethargy, ruffled fur and hunched posture were monitored. Mice displaying signs of toxoplasmosis with ~20% loss of starting weight were euthanised according to protocols approved by the James Cook University Animal Ethics Committee.

*Plasmodium berghei* ANKA parasites were propagated in Swiss and C57BL/6 mice. All procedures involving the infection of mice with *P. berghei* ANKA strain were approved by the Australian National University Animal Experimentation Ethics Committee.

**Construction of plasmids and parasite strains.** To generate a vector for the 3′ replacement of the *Tg*NPT1 with a × 1 hemagglutinin (HA) tag, the 3′ region of *Tg*NPT1 was amplified with primers 1 and 2 (listed in Supplementary Table 1), using TATi/Δku80 parasite genomic DNA as template. The resulting product was digested with *Bam*HI and *Avr*II and ligated into the *Bgl*II and *Avr*II sites of pgCH (a vector that enables fusion of the 3′ flank of interest to a × 1 HA tag). The resultant vector (termed *Tg*NPT1 3′rep in pgCH) was linearized with *Nsi*I, transfected into TATi/Δku80 parasites, and selected on chloramphenicol as described[40]. The resultant parasite strain was termed *Tg*NPT1-HA. A clonal line of *Tg*NPT1-HA parasites was screened for correct integration of the HA cassette. Genomic DNA was extracted from parasites and used as a template for PCR, using primers 3 and 4, which will only detect a PCR product in the native locus, and primers 3 and 5, which will only detect a PCR product in the modified locus (Supplementary Fig. 2a,b).

To replace the *Tg*NPT1 promoter with an ATc-regulated promoter, a 3′ flank of *Tg*NPT1 was amplified with primers 6 and 7, using TATi/Δku80 parasite genomic DNA as template. The resulting PCR product was digested with *Xma*I and *Not*I and ligated into equivalent sites of the pPR2-HA$_3$ vector[41] to produce the vector pPR2-HA$_3$(*Tg*NPT1 3′ flank). Next, the 5′ flank of *Tg*NPT1 was amplified with primers 8 and 9, using TATi/Δku80 parasite genomic DNA as template. The resulting product was digested with *Pac*I and *Fse*I, and ligated into the equivalent sites of the pPR2-HA$_3$(*Tg*NPT1 3′ flank) vector to produce a vector termed pPR2-HA$_3$(*Tg*NPT1 knock-in). This vector was linearized with *Pac*I and transfected into TATi/Δku80 parasites. Although subsequent selection on pyrimethamine yielded drug-resistant parasites, none of the resulting clones were successfully integrated at the *Tg*NPT1 locus. The pPR2-HA$_3$(*Tg*NPT1 knock-in) vector adds a × 3-HA tag to the N-terminus of *Tg*NPT1. We reasoned that this N-terminal HA tag might be interfering with functioning and/or targeting of *Tg*NPT1. We therefore generated a vector where we deleted the × 3-HA tag from the pPR2-HA$_3$(*Tg*NPT1 knockin) vector by digesting this with *Nhe*I and *Sma*I (*Nhe*I and *Sma*I flank either side of the HA tag). The digested vector was then treated with the Klenow fragment of DNA polymerase in the presence of dNTPs to fill in the 5′ overhang of *Nhe*I and create blunt ends at both termini of the plasmid. The vector was then re-ligated. The resultant vector was termed pPR2(*Tg*NPT1 knock-in). This vector was transfected into TATi/Δku80 strain

parasites and selected on pyrimethamine as described[40]. The resultant parasite strain was termed i*Tg*NPT1.

Clones of the i*Tg*NPT1 strain were screened for successful integration of the ATc-regulated promoter upstream of the *Tg*NPT1 start codon. Genomic DNA was extracted from clonal parasites and used as template for PCRs with primers 3 and 10 (to detect presence of the native gene) and primers 10 and 11 (to detect presence of the ATc-regulated promoter at the 5′ end of the modified *Tg*NPT1 gene; Supplementary Fig. 2c,d).

To detect *Tg*NPT1 protein in the i*Tg*NPT1 parasite strain, a × 1HA tag was integrated into the i*Tg*NPT1 locus. To do this, the *Tg*NPT1 3′rep in pgCH vector was digested with *Nsi*I and transfected into i*Tg*NPT1 parasites (Supplementary Fig. 2c). Parasites were selected on chloramphenicol then cloned out. We termed the resultant parasite strain i*Tg*NPT1-HA.

To enable the quantitative measurement of growth in i*Tg*NPT1 strain parasites, a vector carrying a tandem dimeric Tomato red fluorescent protein (tdTomato) construct[42,43] was transfected into i*Tg*NPT1 strain parasites. Three days after transfection, tdTomato expressing parasites were selected by fluorescence-activated cell sorting (FACS) on a FACS Aria II cell sorter (BD Biosciences). After one further round of FACS-based selection, parasites were cloned to produce a parasite strain termed i*Tg*NPT1/Tomato. These parasites were used in fluorescence growth assays, as described previously[18,19], to determine growth phenotypes in various media. Briefly, 2000 parasites expressing tdTomato were inoculated into wells of an optical bottom 96-well plate containing confluent HFF host cells. Plates were incubated at 37 °C in a 5% CO$_2$ incubator and the fluorescence measured every day for 5–10 days in a FluoStar Optima fluorescence plate reader (BMG Labtech). Parasite growth curves were plotted using Prism 6. For experiments examining concentration-dependence of parasite growth on arginine and lysine, growth was plotted at a time point (typically 4–5 days) when parasites cultured in optimal medium conditions had reached mid-logarithmic stage.

To determine whether the growth phenotypes exhibited by the i*Tg*NPT1/ Tomato strain were due solely to knockdown of *Tg*NPT1, the i*Tg*NPT1/Tomato parasite strain was complemented with constitutively-expressed *Tg*NPT1. To do this, the entire open reading frame of *Tg*NPT1 was PCR amplified with primers 2 and 12, using complementary DNA (cDNA) from wild type RH strain parasites as template. The resultant PCR product was digested with *Bam*HI and *Avr*II and ligated into the *Bgl*II and *Avr*II sites of the vector pUgCTH$_3$. The resulting vector was termed c*Tg*NPT1/pUgCTH$_3$, and expresses *Tg*NPT1 from the constitutive α-tubulin promoter, with the resultant *Tg*NPT1 protein containing a C-terminal × 3HA tag. This vector also has a flank of the non-essential *Tg*UPRT gene[44] to enable integration of the c*Tg*NPT1/pUgCTH$_3$ vector into the UPRT locus by homologous recombination. This vector was linearized with *Mfe*I, transfected into i*Tg*NPT1/Tomato parasites, selected on chloramphenicol, and subsequently cloned. The resultant parasite strain was termed i*Tg*NPT1/c*Tg*NPT1.

To perform a direct knockout of *Tg*NPT1, a 3′ flank of the *Tg*NPT1 locus downstream of the stop codon was amplified using primers 13 and 14, with genomic DNA from TATi/Δku80 strain parasites used as template. The resultant product was digested with *Spe*I and *Not*I, and ligated into the *Avr*II and *Not*I sites of pPR2-HA$_3$ to produce a vector termed pPR2(*Tg*NPT1 3′flank). Next, a 5′ flank of *Tg*NPT1 upstream of the start codon was amplified with primers 8 and 9, using TATi/Δku80 strain genomic DNA as template. The resulting product was digested with *Pac*I and *Fse*I, and ligated into the equivalent sites of pPR2(*Tg*NPT1 3′flank). This produced a vector termed pPR2(Δ*npt1*). This vector was linearized with *Pac*I, transfected into TATi/Δku80 strain parasites and selected on pyrimethamine, all the while being cultured in RPMI medium. Drug-resistant parasites were cloned by serial dilution, and clones were screened for disruption of the *Tg*NPT1 locus using one set of primers that only gave a PCR product if the native *Tg*NPT1 locus was present (primers 2 and 12), and a second set of primers that only gave a PCR product if the *Tg*NPT1 gene was replaced with the pyrimethamine selection cassette (primers 15 and 16) (Supplementary Fig. 2e,f). The resultant parasite strain was termed Δ*npt1*.

To enable quantitative measurement of growth in Δ*npt1* strain parasites by fluorescent growth assays, a tdTomato construct[42,43] was transfected into Δ*npt1* strain parasites and selected by FACS (described above). This produced a parasite strain that we termed Δ*npt1*/Tomato.

To determine whether the growth phenotype observed in Δ*npt1*/Tomato was because of loss of the *Tg*NPT1 gene, the c*Tg*NPT1/pUgCTH$_3$ vector (described above) was transfected into the Δ*npt1*/Tomato parasites. After selection on chloramphenicol, parasites were cloned and *Tg*NPT1 expression verified by immunofluorescence assays. The resultant parasite strain was termed Δ*npt1*/tubNPT1. Growth assays confirmed full restoration of parasite growth in DMEM medium (not shown). This strain was used to test whether complementation with *Tg*NPT1 restored [$^{14}$C]Arg uptake and virulence of Δ*npt1* parasites.

To generate a knockout strain of the *P. berghei* NPT1 gene, we transfected a *Pb*NPT1 knockout vector (kindly provided by Patricia Baldacci, Institut Pasteur; ref. 14) into *P. berghei* ANKA strain parasites. Transfections were performed as described previously[45]. Briefly, mature schizonts were isolated by density gradient centrifugation and transfected with the *Pb*NPT1 knockout vector using an Amaxa Nucleofector device (set to program U33). Transfected parasites were injected intravenously into three Swiss mice. The mice were given drinking water supplemented with pyrimethamine to select for *Pb*NPT1 knockout (Δ*Pbnpt1*) parasites.

A clonal population of $\Delta Pbnpt1$ knockout parasites was generated by limiting dilution as described elsewhere[46]. Successful integration was confirmed by PCR analysis, using sets of primers that detected either the native gene (primers 17 and 18 for the 5′ region; primers 20 and 21 for the 3′ region) or the disrupted gene (primers 17 and 19 for the 5′ region; primers 21 and 22 for the 3′ region; Supplementary Fig. 2g,h).

To generate a vector for complementary RNA (cRNA) synthesis of TgNPT1-HA for subsequent injection into X. laevis oocytes, the open reading frame of TgNPT1 was PCR amplified from the TgNPT1 cDNA in pUgCTH₃ vector using primers 23 and 24. The resulting product was digested with XmaI and XbaI and ligated into the equivalent sites of the pGEM-He-Juel (pGHJ) vector[47]. This places TgNPT1 downstream of a T7 RNA polymerase promoter to enable cRNA transcription of TgNPT1, and also introduces a C-terminal HA tag onto the resultant protein. The resulting transcript contains 5′ and 3′ untranslated regions of the X. laevis β-globin gene, which facilitate translation of the transcript following injection into oocytes. The resulting vector was termed TgNPT1 in pGHJ-HA.

To generate a vector for cRNA synthesis of PbNPT1, the open reading frame of PbNPT1 was codon-harmonized for expression in X. laevis (sequence available from the authors upon request). The harmonized version of PbNPT1 was synthesized as a gBlock gene fragment (Integrated DNA Technologies), and the open reading frame was amplified using primers 25 and 26. The resultant PCR product was digested with XmaI and AvrII, and ligated into the equivalent sites of TgNPT1 in pGHJ-HA. This places codon-harmonized PbNPT1 downstream of a T7 RNA polymerase promoter to enable cRNA transcription, and fuses a C-terminal HA tag onto the resultant protein. We termed the resultant vector PbNPT1 in pGHJ-HA.

**Immunofluorescence assays and microscopy.** For immunofluorescence assays, parasites were fixed in 3% (v/v) paraformaldehyde in PBS, permeabilised with 0.25% (v/v) Triton X-100 in PBS, and blocked with 2% (w/v) bovine serum albumin. Parasites were incubated in primary and fluorophore-conjugated secondary antibodies for one hour each, before being mounted in FluoroGel mounting medium (Electron Microscopy Sciences). Fluorescence images were acquired on a DeltaVision Elite system (GE Healthcare) using an inverted Olympus IX71 microscope with a × 100 UPlanSApo oil immersion lens (NA 1.40). Images were recorded using a Photometrics CoolSNAP HQ² camera, deconvolved using SoftWoRx Suite 2.0 software, and adjusted for contrast and brightness. All images were processed further with Adobe Illustrator CS6 software. Primary antibodies used in this study were monoclonal rat anti-HA (1:100 dilution; clone 3F10, Roche) and monoclonal mouse anti-TgSAG1 (1:1,000; clone TP3, Abcam, catalogue number ab8313). Secondary antibodies used in this study were anti-mouse AlexaFluor 546 (1:500; Life Technologies, catalogue number A-11030), anti-mouse AlexaFluor 647 (1:500; Life Technologies, catalogue number A-21236), and anti-rat AlexaFluor 488 (1:200; Life Technologies, catalogue number A-11006).

**SDS-PAGE and western blotting.** Proteins were separated by SDS-PAGE using NuPAGE 12% Bis-Tris gels according to the manufacturer's instructions (Thermo Scientific). Proteins were transferred to nitrocellulose membranes using an XCell II blot module according to the manufacturer's instructions (Thermo Scientific). Membranes were blocked in Blotto (4% (w/v) skim milk powder in Tris-buffered saline (TBS)), before being incubated in primary antibodies for 1 h. Membrane was washed twice in Blotto, then twice in Tween-TBS (T-TBS; 0.05% (v/v) Tween 20 in TBS), before being exposed to secondary antibodies for a further hour. Membrane were washed twice in Blotto and twice in T-TBS, before being incubated in ECL plus western blotting substrate (Thermo Scientific) and exposed to film. Primary antibodies used in this study were mouse anti-GRA8 (1:10,000 dilution; a kind gift from Gary Ward, U. Vermont[48]) and rat anti-HA (1:100 dilution; clone 3F10, Roche). Horseradish peroxidase (HRP)-conjugated secondary antibodies (Santa Cruz Biotechnology, catalogue numbers sc-2005 and sc-2006) were diluted between 1:5,000–1:10,000.

**Oocyte expression and flux analysis.** Xenopus laevis oocyte experiments were performed as described previously[20]. Briefly, TgNPT1-HA and PbNPT1-HA complementary RNAs (cRNAs) were synthesized using the mMessage mMachine T7 transcription kit (Life Technologies) according to the manufacturer's instructions, with the exception that 1 mM nucleotides was used to increase the yield of cRNA. Harvested oocytes were defolliculated with 2 mg ml$^{-1}$ collagenase D (from Clostridium histolyticum; 0.3 U mg$^{-1}$; Roche) in OR$^{2-}$ buffer (82.5 mM NaCl, 2.5 mM KCl, 1.0 mM MgCl₂, 1.0 mM Na₂HPO₄, 5 mM HEPES, pH 7.8). Collagenase was removed by washing oocytes several times with OR$^{2-}$ buffer and incubated overnight at 18 °C in OR$^{2+}$ buffer (OR$^{2-}$ supplemented with 1 mM CaCl₂). The next day, oocytes were micro-injected with 50 nl of cRNA in water at a concentration of 1 μg μl$^{-1}$, using a Nanoject II Auto Nanoliter microinjection device (Drummond Scientific Company), or were not injected. To determine whether HA-tagged TgNPT1 and PbNPT1 localized to the plasma membrane of oocytes, cRNA-injected oocytes were subjected to surface biotinylation to selectively label proteins on the plasma membrane. To do this, oocytes were washed three times with ice cold PBS, pH 8.0. Oocytes were then

incubated with 0.5 mg ml$^{-1}$ EZ-Link sulfo-NHS-SS-Biotin (Thermo Scientific) in PBS for 30 min at room temperature. The reagent was removed by washing oocytes five times with ice-cold PBS, pH 8.0. Subsequently, oocytes were lysed by incubation in lysis buffer (150 mM NaCl, 20 mM Tris-HCl, pH 7.5, 1% Triton X-100 (v/v)) for 1 h on ice. The lysate was centrifuged at 16,000$g$ for 15 min at 4 °C. Lysate supernatant extracted from 5–10 biotin-treated TgNPT1-expressing oocytes, PbNPT1-expressing oocytes, or uninjected oocytes were affinity purified using streptavidin-conjugated agarose beads. Affinity purified proteins (at a loading equivalent of one oocyte per lane) were separated by SDS–PAGE and subjected to western blotting with anti-HA antibodies.

Oocyte flux assays were performed as described previously[49]. Briefly, in each experiment, 7–10 oocytes from a single frog (cRNA or uninjected controls) were washed three times in ND96 solution (96 mM NaCl, 2 mM KCl, 1.8 mM MgCl₂, 1 mM CaCl₂ and 5 mM HEPES hemisodium, pH 7.4). The oocytes were incubated for 0–30 min in 100 μl ND96 supplemented with 0.1–0.4 μCi ml$^{-1}$ [¹⁴C]Arg or 0.1 μCi ml$^{-1}$ [¹⁴C]Lys. For the experiments in Figs 3a and 4a,b, and Supplementary Figs 5b,e,f, 100 μM unlabelled arginine or lysine was added to the uptake solution. Where appropriate, uptake assays were performed in the presence of a 1 mM concentration of various unlabelled amino acids. For the ion replacement experiments, Na$^+$ was replaced with N-methyl-D-glucamine (NMDG), Cl$^-$ replaced with gluconate, and K$^+$, Mg$^{2+}$ and Ca$^{2+}$ replaced with Na$^+$. All buffers had a final osmolarity of ~200 mOsm l$^{-1}$. Preliminary experiments indicated that several amino acids, including methionine, either inhibited the uptake of radiolabelled arginine by uninjected oocytes, or inhibited the non-specific binding of radiolabelled arginine to oocytes (Fig. 3b). The uptake of [¹⁴C]Arg and [¹⁴C]Lys into oocytes expressing PbNPT1 was therefore measured in the presence of 125–500 μM unlabelled methionine in order to minimize the 'background signal'. Uptake of radiolabelled amino acid was terminated by adding a large volume of ice cold ND96, and the extracellular solution was then removed by three washes in ice cold ND96. Oocytes were separated into scintillation vials (a single oocyte per vial) and lysed in 200 μl of 10% (w/v) SDS solution before the addition of Microscint fluid (Perkin Elmer). The radioactivity in each sample was measured using a Perkin Elmer scintillation counter. In each experiment the data obtained for each condition tested was averaged from 7 to 10 oocytes. Experiments were performed in triplicate, with each of the three experiments using oocytes harvested from different X. laevis frogs. X. laevis handling and experimentation procedures were approved by the Australian National University Animal Experimentation Ethics Committee.

**Electrophysiological recordings.** The two-electrode voltage clamp technique was used as described previously[21]. Briefly, micropipettes were pulled using 1.5 mm diameter borosilicate glass capillaries and backfilled with 2 M KCl. For voltage clamp recordings, two electrodes were connected to the head stages of an Axoclamp-2B amplifier (Axon Instruments). The experimental bath was grounded with a chloride-treated silver wire coated with 3% (w/v) agarose dissolved in ND96. Oocytes were voltage-clamped at − 50 mV and placed in a constant flow of ND96 solution. Once the current had stabilised the oocytes were exposed to ND96 solution containing 5 mM arginine. For the experiments giving rise to Fig. 4d, the arginine-induced currents were measured in oocytes exposed to media in which various ions were replaced; Na$^+$ was replaced with NMDG, Cl$^-$ was replaced with gluconate, and K$^+$, Mg$^{2+}$ and Ca$^{2+}$ were replaced with Na$^+$. All buffers had a final osmolarity of ~200 mOsm l$^{-1}$. In all cases, the pH was adjusted to 7.4. For the experiment giving rise to Fig. 4e, solutions of varying pH were prepared using a combination of the buffering agents MES (pK$_a$ = 6.2) and Tris–HCl (pK$_a$ = 8.3) at different ratios, giving a combined final concentration of 5 mM and keeping the osmolarity of the solution constant. Current measurements were performed at 300 ms frequency and were monitored using Axon Axoclamp software.

**Parasite flux assays.** Freshly egressed T. gondii parasites were suspended at 37 °C and at a cell density of ~1 × 10$^8$ cells ml$^{-1}$ in Dulbecco's phosphate-buffered saline (PBS; Sigma), pH 7.4, supplemented with 25 mM glucose, and containing either 0.1 μCi ml$^{-1}$ [¹⁴C]Arg (with a specific activity of 312 mCi mmol$^{-1}$) or 0.1 μCi ml$^{-1}$ [¹⁴C]Lys (specific activity 326 mCi mmol$^{-1}$) together with unlabelled amino acids (where specified). Aliquots (200 μl) were sampled at predetermined time points, and centrifuged at 12,000$g$ for 30 s through 250 μl of an oil mix comprising of 84% (v/v) PM-125 silicon fluid (Clearco) and 16% (v/v) light mineral oil, similar to a protocol described previously[50]. The supernatant solution above the oil layer was aspirated, the tubes were washed three times in H₂O, and the oil layer was then aspirated, leaving the parasite pellet. The pellets were lysed in 0.1% (v/v) Triton X-100 in H₂O, then mixed with microscint fluid (Perkin Elmer). Radioactivity was measured using a Perkin Elmer scintillation counter. The uptake time-course data were fitted by a single exponential function and the initial rate of amino acid transport was estimated from the initial slope of the fitted line.

Uptake assays in P. berghei parasites were performed using a procedure modified from one described previously for P. falciparum[31]. P. berghei parasites, suspended in PBS supplemented with 11 mM glucose (PBS + glucose), were separated from uninfected blood cells using a VarioMACS separation unit[51,52], then isolated from host erythrocytes by saponin permeabilisation of the erythrocyte and parasitophorous vacuole membranes as described[53]. For experiments

measuring the uptake of [$^{14}$C]Arg or [$^{14}$C]Lys, isolated parasites were washed and resuspended at $\sim 2 \times 10^8$ cells ml$^{-1}$ in PBS + glucose at 37 °C. For experiments measuring the uptake of [1-$^{14}$C]2-DOG (specific activity 55 mCi mmol$^{-1}$), isolated parasites were washed and resuspended in PBS supplemented with 50 μM glucose. In all cases, the uptake assay commenced with the addition of 100 μl PBS + glucose containing 0.2 μCi ml$^{-1}$ of either [$^{14}$C]Arg, [$^{14}$C]Lys or [1-$^{14}$C]2-DOG to 100 μl of the cell suspension.

At predetermined time points, uptake was terminated by centrifuging the parasite suspension through a 300 μl oil mix composed of dibutyl phthalate and dioctyl phthalate (5:4) layered on 30 μl of 30% (v/v) perchloric acid in H$_2$O at 17,000g for 1 min. After sedimentation, the aqueous supernatant above the oil mix was aspirated and the residual radioactivity on the walls of the tube was removed by rinsing three times with water. Following the final wash, the oil layer was aspirated, and the parasite pellet was lysed in 0.1% (v/v) Triton X-100 in H$_2$O and centrifuged at 17,000g for 2 min. The supernatant was transferred into a vial containing scintillation fluid, vortexed and radiation was measured using a Perkin Elmer scintillation counter.

For both T. gondii and P. berghei flux assays, the unincorporated radiolabel trapped in the extracellular solution between parasites as they were centrifuged through the oil layer was estimated by rapid sampling (<15 s) following the addition of radiolabelled substrate to the cell suspension. In the [$^{14}$C]Arg and [$^{14}$C]Lys uptake experiments, a 5–10 mM concentration of an unlabelled form of the amino acid was added at the same time as radiolabel to slow the uptake of radiolabelled substrate. In the [$^{14}$C]2-DOG uptake experiments a 5 mM concentration of unlabelled glucose was added, for the same purpose. In each uptake experiment the amount of radiolabel measured in the extracellular solution was subtracted from the total radiolabel in the cell pellet for all samples, to give the amount of radiolabel incorporated into the parasite.

**Quantification of microgametogenesis.** To determine whether ΔPbnpt1 parasites were deficient in formation of male gametes (as reported previously[14]), microgametogenesis was induced in WT and ΔPbnpt1 parasites by a drop in temperature and the addition of xanthurenic acid. The extent of [$^3$H]hypoxanthine incorporation into DNA that is newly synthesized upon exflagellation of the microgametocytes was measured as described previously[54]. Briefly, wild type and ΔPbnpt1 parasite cultures were magnet purified using a VarioMACS separation unit and resuspended in hypoxanthine-free RPMI-HEPES (RPMI supplemented with 20 mM HEPES and 4 mM sodium bicarbonate, pH 8.0). Aliquots of purified parasites were transferred into wells of a 96-well plate to a final volume of 200 μl. 25 μl of gametocyte activating media (RPMI-HEPES supplemented with 100 μM xanthurenic acid and 16 μCi ml$^{-1}$ [$^3$H]hypoxanthine) was added to each well. The plate was incubated for 10 min at room temperature, during which time competent male gametocytes will undergo three rounds of genome replication followed by exflagellation, then frozen at −20 °C. The plate was thawed to lyse the cells, and radiolabelled DNA transferred to filters using a Filtermate Universal Cell Harvester (Perkin Elmer). The samples were read using a MicroBeta plate counter (Perkin Elmer). This provides a quantitative measure for microgametogenesis.

**Alignments of apicomplexan NPT proteins.** NPT-family proteins from T. gondii, P. berghei and P. falciparum were identified in www.toxodb.org and www.plasmodb.org. Protein sequences were aligned using ClustalX 2.1 (ref. 55) and a visual output of the sequence alignment was generated using the BoxShade webserver (http://www.ch.embnet.org/software/BOX_form.html). Accession numbers of the sequences were as follows: TGME49_215490 (TgNPT1), TGME49_320020, PBANKA_020830 (PbNPT1), PBANKA_081570, and PF3D7_0104800 (PfNPT1). A topology model to predict transmembrane domains of TgNPT1 was constructed using PHOBIUS (ref. 56).

**Statistical analyses.** Paired and unpaired two-tailed Student's t tests and analyses of variance (ANOVAs) were performed, as appropriate, using Prism. The significance level was set to 0.05; where $P > 0.05$, comparisons were considered not significant.

**Data availability.** Gene sequences for NPT-family proteins from T. gondii, P. berghei and P. falciparum were identified in www.toxodb.org and www.plasmodb.org with accession codes TGME49_215490 (TgNPT1), TGME49_320020, PBANKA_020830 (PbNPT1), PBANKA_081570, and PF3D7_0104800 (PfNPT1). All other data supporting the findings of this study are available within the article and its Supplementary information files and from the corresponding authors on reasonable request.

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

## Acknowledgements

We are grateful to Ben Corry for comments on the manuscript, to Kathryn Parker and Markus Winterberg for helpful discussions, to Patricia Baldacci, Gary Ward, Lilach Sheiner, Boris Striepen, Nick Katris and Chris Tonkin for sharing reagents, to Harpreet Vohra and Azadeh Seidi for technical assistance, and to the students of the 2014 Biology of Parasitism course (Woods Hole, MA) for preliminary studies. This work was supported by a Discovery Grant from the Australian Research Council (ARC) to K.K., G.v.D., S.B. and I.C. (DP150102883) and an NHMRC Project Grant to K.K. (525428). G.v.D. was supported by an ARC QEII Fellowship (DP110103144).

## Author contributions

E.R., K.K. and G.v.D designed research; E.R., S.H., C.M., Y.C., I.C. and G.v.D performed research; E.R., S.H., C.M., S.F., S.B., K.K. and G.v.D. analysed results; N.S. contributed new reagents/analytic tools; E.R., K.K. and G.v.D. wrote the manuscript with contributions from all the authors.

## Additional information

**Competing financial interests:** The authors declare no competing financial interests.

