## [Peer Review File · Nature Communications]

Reviewer #1 (Remarks to the Author)

This manuscript describes the elegant biochemical characterization of the TgNPT1 and PbNPT1 arginine transporters from *Toxoplasma gondii* and *Plasmodium berghei*, respectively. These organisms are apicomplexan parasites and etiologic agents of important and impactful diseases around the world. TgNPT1 and PbNPT1 are the first apicomplexan amino acid transporters to be identified at the molecular level, obviously an original accomplishment, but amino acid transporters, including those more-or-less specific for arginine, have been identified in other eukaryotic systems, including mammals and other protozoan parasites. Both TgNPT1 and PbNPT1 are, according to the authors, members of an apicomplexan-specific putative transporter family, but the degree to which TgNPT1 and PbNPT1 are structurally different from other eukaryotic members of the cationic amino acid transporter family is not further elaborated. Thus, the significance and novelty of these reported analyses are not completely transparent.

Overall, the manuscript is very well-written, the data are persuasive and of high quality, and the conclusions are substantiated by the findings. PbNPT1 was initially characterized, as part of a large functional screen of orphan membrane transporters, by another group (reference #13) and determined to be an essential determinant of gametogenesis in *P. berghei*. In this submission, Rajendran and coworkers assigned a biochemical function to PbNPT1, that of a cationic amino acid transporter, and did so via a route that involved first the biochemical and genetic characterization of its homolog, TgNPT1, which is an arginine-specific, or at least an arginine-preferring, amino acid transporter. The rationale for first characterizing TgNPT1 is that *T. gondii* is, by far, the more pliant experimental system for studying the biology and biochemistry of apicomplexan parasites. The initial biochemical characterization of TgNPT1 involved the construction of anhydrotetracycline repressible, as well as knockout, TgNPT1, lines, and genetic repression or elimination of TgNPT1 conferred a conditionally lethal phenotype in DMEM but not RPMI medium, which, by a series of inferences and hypothesis-testing experiments, was ultimately shown to be due to differences in the arginine composition of the two media formulations. The biochemical characterization of the TgNPT1 in *Xenopus* oocytes that ensued was straightforward and the protein shown to be arginine-selective. Since TgNPT1 was a known homolog to PbNPT1, and the phenotypic consequences of genetic lesion in PbNPT1 was known, it was easy to infer, and then to experimentally ascribe, a biochemical function to the malarial homolog, although its ligand selectivity was somewhat more promiscuous than that for TgNPT1.

I have a couple of concerns. How unique is the characterization of yet another cationic amino acid transporter? Is this protein profoundly different structurally from its functional counterparts in other eukaryotic systems? This matter is worthy of additional discussion. Second, it is remarkable that a <3-fold difference in arginine composition in RPMI-based and DME-based media can have such a profound all-or-nothing effect on growth of *T. gondii* when TgNPT1 activity is down-regulated or genetically abrogated. Comments directed to this issue would also be illuminating.

Reviewer #2 (Remarks to the Author)

The phylum Apicomplexa comprises parasites like *Toxoplasma gondii* and *Plasmodium* spp. responsible for the diseases toxoplasmosis and malaria, respectively. The authors show that members of a previously non-characterized, Apicomplexa-specific gene family mediate transport of cationic amino acids both in heterologous expression systems and in the parasite. Furthermore the transporters were shown to be essential for parasite survival and virulence. This is novel and very exciting data, also interesting for a wider audience.

A number of additional controls have to be included and some editorial changes are suggested.

Major points

Fig. 1b. To confirm localization at the parasite plasma membrane, the Pearson correlation

coefficient should be given, alternatively use super-resolution microscopy.

Fig. 3a and Fig. 6b. Uptake has to be given as pmol oocyte⁻¹ min⁻¹ and not as cpm. Alternatively, it can be given in %, but if so, the absolute uptake rates should be given for the control (in Fig. legend), e.g. 100 % corresponds to X pmol min⁻¹ oocyte⁻¹.

Fig. 4a/b. Absolute uptake rates should be given for the control, e.g. 100 % corresponds to X pmol min⁻¹ 10^x parasites⁻¹.

Fig 4a. Arg uptake of wild type in the presence of lysine is missing and has to be included.

Fig. 4b. Concentration of lysine used for the experiment has to be given. Please also indicate whether uptake - at such low concentrations - was linear over time.

Fig. 7b. Uptake of a metabolite that is not a substrate of NPT1 (e.g. non-related amino acid or glucose) has to be included to show that cells are exclusively impaired in arg/lys transport.

All experiments with $\Delta npt1$ have to include the "addback" as control, i.e. Figures 4, 5, 7

All figure legends (and where appropriate also in the text) should indicate concentrations of (labelled) amino acids used for the transport assays.

e.g. line 663: "[¹⁴C]Arg uptake into..." → **289 nM** [¹⁴C]Arg uptake into

This is especially confusing in Figure 4, i.e. in Fig. 4a, 40 μ M [¹⁴C]Arg was used, whereas the concentration of lysine in Fig 4b, seems not to contain any unlabelled lysine (and thus concentration is probably in the nM range).

177, 308. Uptake into oocytes was measured over 30 min... It did not become entirely clear whether these are "true" uptake rates or end-point analyses. Calculating "true" uptake rates (based on several time points measured) would be strongly preferred. If this are end-point analyses (30 min), data for linearity of uptake over time has to be provided (e.g. as Suppl.Fig.).

349. Hypoargininaemia is an interesting phenomenon, however, this is not well connected to the text, especially as the phenomenon seems attributed to host arginase activity.

General 1: The data do not contain any information on the transport mechanism e.g. driving force, co-transported ions, etc. When working with oocytes two-electrode voltage clamp analyses are obvious and could provide these data. In case substrate-induced currents are too small to perform such analyses, the authors should at least mention this.

Also export can be easily measured by injecting radiolabeled amino acids into the oocytes. Though export may not be physiologically relevant, these results would allow drawing additional conclusions on the mechanism of transport. As this is a member of a novel protein family, this would be valuable information that would strengthen the biochemical characterization of NPT1.

General 2: For readers not in the field, it would be interesting to get a wider perspective and discussion.

For example:

- How does the transport selectivity, transport mechanism etc. compare with arginine/lysine

transporters in other organisms like human, yeast, other protozoan parasites, etc.

- Though NPTs are specific for Apicomplexa, homologs of amino acid transporters from other kingdoms seem to be present in Apicomplexa genomes and may, in addition to other NPTs, contribute to amino acid uptake.

-

Minor points

62-72. This could be considerably shortened e.g. by combining the findings in both parasite species, rather than doing this sequentially and by removing parts of the sentences that are not adding e.g. "we go on to show that".

92-96. Molecular mass (Western blot).

"Western blotting revealed that TgNPT1-HA has a molecular mass of ~40 kDa, below the calculated mass of 58 kDa (Fig. 1a). The reason for this discrepancy is unclear."

Please rephrase: The reason for this discrepancy was not further investigated.

As mentioned, having an apparent smaller molecular mass on SDS PAGE is a common feature of membrane proteins, but cleavage can not be excluded. Whether N-terminal cleavage occurs, could be assessed e.g. by adding a tag at the N-terminus. The information that the author tried adding an N-terminal tag, but could not get respective clones, can only be found in the cloning procedure given in the Suppl. M&M 144. This information could be added to the main text to clarify why this could not be assessed.

97. ...TgNPT1-HA localizes to the periphery of the parasite, as well as to some internal structures (Fig.1b).

Please provide information on potential identity of structures (partially also seen in the SAG1 control) and/or physiological role of transporters in internal compartments.

109. Please rephrase: revealed there to be no detectable.... → revealed no detectable

180, Fig 3A and 6B. Transport in the absence of competitors should be labelled as control, i.e. in the absence of unlabelled amino acids (like in Fig 4b), and not as "carrier-free".

303-339, PbNPT1. After presenting the results of *T. gondii*, essentially the same experiments are repeated with *P. berghei*. These results are certainly essential. However, the authors should considerably shorten the *P. berghei* chapter. i.e. the descriptions of the experiments should be more concise. Details can be included in figure legends.

330. Also include the published finding that PbNPT1 is localized at the plasma membrane

335. "functionally isolated from their host erythrocyte"... Please rephrase.

440. "suggested that methionine could inhibit". Rephrase → suggested that several amino acids including methionine could inhibit.

Can the authors exclude that adding methionine inhibits endogenous transport systems or simply reduced unspecific binding?

630. immunofluorescence reveals co-localization. Rephrase: reveals partial co-localization

Please replace lab slang in Supp. Information e.g.

39, and several other lines. "predicted to give a band" → predicted to amplify a fragment

147. "plates were grown" → plates were incubated or cells were grown.

Was in any experiment NPT1-mediated transport in the absence of a HA tag compared with NPT1-HA that is generally used? Not sure whether I missed this.

General comment: It is not common to do competition experiments at substrate concentrations so far below the K_m of the transporter and use competing substrates at >3000-fold excess.

Competition experiments at V_{max} or around K_m are preferred for these analyses. As both background and NPT1-mediated uptake seem to increase linearly at concentrations below K_m , I can not see any reason why the experiments were not performed using concentrations in the physiological range.

As background stays reasonably low at these concentrations, I do not expect changes in the outcome. Nevertheless, changing this in future experiments is highly recommended.

Reviewer #3 (Remarks to the Author)

Our understanding and exploitation of solute transporters in apicomplexan parasites has been hindered by a lack of functional information about these important proteins. Putative transporters that show little or no homology to previously characterised transporters make attempts to identify and demonstrate function extremely difficult. Thus, the author's efforts here are worthy of some praise and should be of interest to the readership of Nature Communications. The findings are novel and are supported by the data, which are generally of a high standard, contain correct controls and are analysed appropriately (including statistically). However, I do have a number of suggestions that the authors might consider to improve the paper.

1. Abstract. Line 28. Maybe "characterises" rather than "identifies" since a number of putative amino acid transporters have been identified previously.

2. Introduction. Line 41. I disagree with this sentence as characterised transporters also have equivalents in plants and fungi and in some cases are absent in mammals. For example, the Plasmodium and Toxoplasma calcium hydrogen exchangers have been characterised, which do not have orthologues in mammalian cells (PMID: 23468629), and the functionally characterised Plasmodium vacuolar iron transporter also lacks mammalian orthologues (PMID: 26786069).

3. General. Please could the authors comment on the physiological concentrations of cationic amino acids that these parasites are likely to encounter?

4. General. I may have missed the section but it's not always clear which statistical test has been used to derive P values, please can this be clarified?

5. Results. Line 117. Here it is stated that 30 mins was used for measuring uptake (and this follows through to PbNPT1 oocyte studies). Why was this time point chosen? Were time courses undertaken? Is uptake linear over this period? As I'm sure the authors are aware, this is important to establish when undertaking concentration-dependence experiments (at both high and low concentrations of unlabelled solute).

6. Results. Section from Line 171. Did the authors consider the use of L-arginine analogues to probe the selectivity of the transporter further?

7. Results. Section from Line 171. Can the authors comment further on the transport mechanism? Facilitative or secondary active? Were additional experiments undertaken to shed light on this?

8. Results. Section from Line 196. Can the authors speculate on the importance of the alternate arginine pathway at physiological arginine concentrations, if likely to be different from those used (see point 3 also)?

9. Results. Line 346. PfNPT1 is mentioned at this point as the likely underlying mechanism for cationic amino acid uptake reported recently in *P. falciparum* by some of the authors of this study. Since this particular transporter was not characterised here perhaps the authors could provide some comment as to the likelihood that the Pb and PfNPT1 have similar function e.g. is there sequence similarity between PbNPT1 and PfNPT1 - the sequence could be added to Supp Fig 1?

10. Results. Line 370. Same as point 1 above.

11. Results. Line 372-374. The authors note that NPTs are they are attractive targets, since they lack homologues in humans. This is only one factor of many that make a target attractive. Perhaps the authors could expand on this and/or find an appropriate reference.

12. Methods. Line 456. What were the specific activities of radio-labelled arginine and lysine?

13. Methods. Line 471. [14C]Arg rather than [14C]arginine for consistency and the same for Lys.

14. Figure 4. Although mentioned in the methods section and main text that initial rates of uptake were determined by fitting an exponential function, it is not completely clear that this is the case for the data presented in Figure 4b. Please could this be clarified (e.g. altering the y axis label and figure legend text)? If this is not the case, some of the data could be explained by inhibition of metabolism rather than transport since the parasite are maintained in glucose and thus are likely ATP replete and metabolically active. Line 221: Fig 4a rather than Fig 3a? Also, the transport data in Figure 4 are presented as relative uptake - could the authors provide actual uptake data/calculated initial uptake values for WT and deltaNPT1 parasites to act as a reference point for readers. This could be added to the figure legend.

15. Supp Figure 4. I may have missed this but what was the protein load in this and other westerns? Also Supp Fig 4a: it is very difficult to see if there is even a control image present in the version I have.

In addressing the comments made by the reviewers we have carried out additional experiments, and have made other changes to the manuscript as requested (as outlined in red below, and in red in the revised manuscript). Additionally, we have made numerous editorial changes to the manuscript, all outlined in red in the revised manuscript. The major experimental changes to the manuscript are as follows:

1. As requested by Reviewers 2 and 3 (and emphasised by the editor), we have undertaken a detailed mechanistic study of the *TgNPT1* transporter. Using a combination of [¹⁴C]arginine uptake assays and the two-electrode voltage clamp electrophysiology technique, we have demonstrated that arginine transport by *TgNPT1* is electrogenic, ion-independent and pH-dependent (Fig. 4).
2. In response to comments made by Reviewers 2 and 3, we have re-measured [¹⁴C]arginine and [¹⁴C]lysine uptake in wild type and $\Delta npt1$ *T. gondii* parasites during the linear phase of uptake (Fig. 5a-b). These experiments support our initial conclusions that arginine uptake is impaired in $\Delta npt1$ parasites, and that the *TgNPT1*-independent uptake of arginine can be inhibited by lysine, suggesting the presence of a general cationic amino acid transporter in these parasites.
3. As requested by Reviewers 2 and 3, we provide evidence that [¹⁴C]arginine and/or [¹⁴C]lysine uptake in *TgNPT1* and *PbNPT1*-expressing oocytes were measured during the linear phase of the uptake time course (Supplementary Fig. 4).
4. In response to comments from Reviewer 2, we now include additional 'addback' (i.e. complementation) experiments, demonstrating that [¹⁴C]arginine uptake (Fig. 5a) and parasite virulence (Fig. 7) are restored upon the constitutive expression of *TgNPT1* in $\Delta npt1$ parasites. These experiments highlight the importance of *TgNPT1* for arginine uptake and virulence in *T. gondii*. Due to a paucity of selectable markers in *Plasmodium berghei*, we were unable to perform the addback experiment in the $\Delta Pbnp1$ strain.
5. In response to a suggestion by Reviewer 2, we demonstrate that uptake and metabolism of the glucose analogue 2-deoxyglucose is not impaired in $\Delta Pbnp1$ parasites (Fig. 9). This supports our hypothesis that the defects in arginine and lysine uptake result specifically from the loss of the *PbNPT1*, and are not a consequence of decreased parasite viability.

Reviewers' comments:

Reviewer #1 (Remarks to the Author):

1. Both *TgNPT1* and *PbNPT1* are, according to the authors, members of an apicomplexan-specific putative transporter family, but the degree to which *TgNPT1* and *PbNPT1* are structurally different from other eukaryotic members of the cationic amino acid transporter family is not further elaborated. Thus, the significance and novelty of these reported analyses are not completely transparent ... How unique is the characterization of yet another cationic amino acid transporter? Is this protein profoundly different structurally from its functional counterparts in other eukaryotic

systems? This matter is worthy of additional discussion.

To highlight the novelty of our findings, we have included the following text in the final paragraph of the Results and Discussion (Lines 435-443 of the revised manuscript):

“Previously characterized arginine/cationic amino acid transporters from eukaryotes belong to the SLC7 or amino acid/auxin permease (AAP) families of transporters (e.g. refs 25, 30). These are fundamentally different, at both primary sequence and structural levels, to the major facilitator superfamily class to which *TgNPT1* and *PbNPT1* belong³⁹. This suggests that apicomplexans have evolved cationic amino acid transporters independently of other major eukaryotic lineages. Apicomplexans lack SLC7 homologues, but harbour homologues of AAP family transporters¹², although the functions of these remain to be elucidated.”

NB: The final sentence addresses a comment from Reviewer 2.

2. It is remarkable that a <3-fold difference in arginine composition in RPMI-based and DME-based media can have such a profound all-or-nothing effect on growth of *T. gondii* when *TgNPT1* activity is down-regulated or genetically abrogated. Comments directed to this issue would also be illuminating.

This is indeed a striking finding, though we should emphasise that it is not simply the <3-fold difference in the arginine concentration in the two media that accounts for our finding, it is the [Arg]:[Lys] ratio that is critical. As is explained in the manuscript (and as is illustrated in what is now Fig. 6 in the revised manuscript), in parasites in which *TgNPT1* activity is down-regulated or genetically abrogated, arginine is taken up via a second transporter that transports both arginine and lysine. Lysine in the medium competes with arginine for uptake via this route (Fig. 5a-b), and it is for this reason that the ratio of [Arg]:[Lys], rather than the absolute concentrations of these amino acids, modulates the growth response. As demonstrated in Fig. 5c, when the ratio of [Arg]:[Lys] is high (~4:1), parasite growth is normal, as there is sufficient Arg entering the parasite. By contrast, when this ratio decreases (<2:1) competition by lysine prevents sufficient arginine entering the parasite via the second, *TgNPT1*-independent transporter, and parasites lacking *TgNPT1* are unable to grow. We note in the text that RPMI has an Arg:Lys ratio of 5.25:1, whereas DME has a ratio of 0.5:1, so there is actually a ~10-fold difference in the ratio between the two media. We have modified the text as follows (Lines 311-315 of the revised manuscript):

“However, when the balance of [Arg]:[Lys] is shifted in the other direction (as is the case in DMEM, in which the [Arg]:[Lys] ratio is 0.5, a 10-fold reduction relative to RPMI), lysine competes with arginine for uptake through the general pathway, and parasites become reliant on *TgNPT1* to take up sufficient arginine to support growth (Fig. 6).”

Reviewer #2 (Remarks to the Author):

1. Fig. 1b. To confirm localization at the parasite plasma membrane, the Pearson correlation coefficient should be given, alternatively use super-resolution microscopy.

As suggested by the reviewer, we have calculated the Pearson's Correlation Coefficient for the TgNPT1-HA and SAG1 (plasma membrane marker) co-localisation. This suggests a high level of correlation between the two, supporting the conclusion that TgNPT1-HA localises to the plasma membrane. We have modified the text in the figure legend as follows (Lines 781-783 of the revised manuscript):

“Immunofluorescence assay of TgNPT1-HA (green) reveals partial colocalization with the plasma membrane marker SAG1 (red) (Pearson's Correlation Coefficient mean \pm SD = 0.81 ± 0.04 , n=6).”

2. Fig. 3a and Fig. 6b. Uptake has to be given as pmol/oocyte/min and not as cpm. Alternatively, it can be given in %, but if so, the absolute uptake rates should be given for the control (in Fig. legend), e.g. 100 % corresponds to X pmol/min/oocyte.

As suggested by the Reviewer, we have converted uptake to fmol/oocyte/30 min in Fig. 3b (previously Fig. 3a) and Fig. 8b (previously Fig. 6b).

3. Fig. 4a/b. Absolute uptake rates should be given for the control, e.g. 100 % corresponds to X pmol/min/ 10^x parasites

As suggested, the uptake data in Fig. 5b (formerly Fig. 4b) is now expressed as pmol/ 10^7 cells/3 min. For Fig. 5a (previous Fig. 4a), we refer to these data in the text in terms of % of control, so have maintained this in the figure. We now state in the figure legend (Lines 875-877) that:

“The mean initial rate of [14 C]Arg uptake in WT parasite measured in the absence of lysine) corresponds to 900 pmol \pm 61 pmol/ 10^7 cells/min (mean \pm SEM; n = 3)/ 10^7 cells/min.”

Additionally, we include the time course of uptake for Fig. 5a as Supplementary Fig. 6a.

4. Fig 4a. Arg uptake of wild type in the presence of lysine is missing and has to be included.

We now include these data (Fig. 5a). As noted in our response to comment 14 by Reviewer 3, we have redone the experiments shown in Fig. 4a (now Fig. 5a) using a more sensitive oil stop method. We also include the 'addback' control (as suggested in comment 7 by Reviewer 2).

5. Fig. 4b. Concentration of lysine used for the experiment has to be given. Please also indicate whether uptake - at such low concentrations - was linear over time.

We have now included the concentration of unlabeled lysine used in these assays. The figure legend in Fig. 5b (previously Fig. 4b) reads (Lines 879-882):

“The uptake of 0.1 μ Ci/ml (307 nM) [14 C]Lys (in the presence of 50 μ M unlabelled lysine) was measured over 3 min (within the initial linear phase of uptake; Supplementary Fig. 6b) in WT (black) and $\Delta npt1$ (grey) parasites, ...”

6. Fig. 7b. Uptake of a metabolite that is not a substrate of NPT1 (e.g. non-related amino acid or glucose) has to be included to show that cells are exclusively impaired in arg/lys transport.

As suggested by the reviewer, we have performed additional experiments to measure the uptake of 2-deoxy-glucose (2-DOG), a glucose analogue that is not a substrate of *PbNPT1*, in WT and $\Delta Pbnpt1$ parasites. We found no difference in [^{14}C]2-DOG incorporation between these parasites. Note that our experimental approach does not distinguish between 2-DOG uptake and subsequent metabolism (phosphorylation by hexokinase), and the rate of incorporation of 2-DOG likely reflects a combination of uptake and metabolism (with the hexokinase-catalysed reaction likely the rate-limiting step). We conclude, therefore, that loss of *PbNPT1* results in a defect in the uptake of Lys and Arg, and is not the result of a loss of parasite viability (as would be observed if 2-DOG incorporation was decreased). We include these data in Fig. 9b and note in the text (Lines 396-403 of the revised manuscript) that:

“The uptake and incorporation of the glucose analogue [^{14}C]2-deoxy-glucose (2-DOG) was similar in WT and $\Delta Pbnpt1$ parasites (Fig. 8b; $P > 0.05$, $n=3$). This indicates that $\Delta Pbnpt1$ parasites remained metabolically active, and suggests that defects in the uptake of cationic amino acids was not the result of general defects in parasite viability. Together, these findings are consistent with *PbNPT1* playing a major and specific role in the uptake of cationic amino acids into the intracellular malaria parasite.”

7. All experiments with $\Delta npt1$ have to include the "addback" as control, i.e. Figures 4, 5, 7.

In our original submission, we demonstrated that growth of NPT1 knockdown parasites in DME medium was restored upon the 'addback' of *TgNPT1* (Fig. 1E). To address the reviewer's comments, we have performed additional experiments that show that:

- Arginine uptake is restored in complemented $\Delta npt1$ parasites (Fig. 5a). Notably, we find that arginine uptake is significantly increased in the complemented cell line. Given that complemented NPT1 is expressed from the constitutive α -tubulin promoter, this is likely the result of *TgNPT1* over-expression, and provides further evidence that *TgNPT1* functions as an arginine transporter.
- Addback of *TgNPT1* into $\Delta npt1$ parasites restores parasite virulence (Fig. 7). This is an important control that implies that loss of virulence in $\Delta npt1$ parasites is due specifically to the loss of this transporter.

We were unable to perform an addback experiment in the $\Delta Pbnpt1$ stain due to the unavailability of selectable markers in this parasite. We note, however, that three separate groups have independently generated a $\Delta Pbnpt1$ strain (our study, Kenthirapalan et al, 2016; Boisson et al, 2011), and all found a similar defect in gametocyte development. Our major finding that *PbNPT1* is a cationic amino acid transporter is supported by both parasite uptake assays, and uptake assays with the heterologously expressed *PbNPT1* transporter in *Xenopus* oocytes.

8. All figure legends (and where appropriate also in the text) should indicate concentrations of (labelled) amino acids used for the transport assays.

e.g. line 663: "[¹⁴C]Arg uptake into..." → 289 nM [¹⁴C]Arg uptake into

This is especially confusing in Figure 4, i.e. in Fig. 4a, 40µM [¹⁴C]Arg was used, whereas the concentration of lysine in Fig 4b, seems not to contain any unlabelled lysine (and thus concentration is probably in the nM range).

We have modified the legend of Fig. 5 (previously Fig. 4) to clarify this. The text reads as follows (Lines 869-882 of the revised manuscript):

“(a) [¹⁴C]Arg uptake in WT, $\Delta npt1$ and $\Delta npt1/tubNPT1$ parasites in the absence (black) and presence (grey) of 80 µM unlabelled lysine, expressed as a percentage of the initial rate of [¹⁴C]Arg uptake in WT parasite measured in the absence of lysine. Uptake was measured in parasites suspended in PBS containing 10 mM glucose, 40 µM unlabelled arginine and 0.1 µCi/ml (289 nM) [¹⁴C]Arg. The initial rates of [¹⁴C]Arg uptake were derived from the initial slopes of the time courses shown in Supplementary Fig. 6a. The mean initial rate of [¹⁴C]Arg uptake in WT parasite measured in the absence of lysine was 900 ± 61 pmol/10⁷ cells/min (mean \pm SEM; n = 3). The data shown represent the mean \pm SEM from three independent experiments (* P < 0.05; ** P < 0.01; *** P < 0.001; Student’s T test). (b) [¹⁴C]Lys uptake in *T. gondii*. The uptake of 0.1 µCi/ml (307 nM) [¹⁴C]Lys (in the presence of 50 µM unlabelled lysine) was measured over 3 min (within the initial linear phase of uptake; Supplementary Fig. 6b) in WT (black) and $\Delta npt1$ (grey) parasites ...”

We have also modified the other figure legends to specify the concentration of substrates that we used for each experiment.

9. 177, 308. Uptake into oocytes was measured over 30 min... It did not become entirely clear whether these are "true" uptake rates or end-point analyses. Calculating "true" uptake rates (based on several time points measured) would be strongly preferred. If these are end-point analyses (30 min), data for linearity of uptake over time has to be provided (e.g. as Suppl.Fig.).

We performed these as end-point assays. To address the reviewer’s point, we performed additional time course uptake assays for [¹⁴C]Arg in *TgNPT1*-expressing oocytes, and [¹⁴C]Arg and [¹⁴C]Lys in *PbNPT1*-expressing oocytes across 60-80 min. These experiments indicated that uptake of all these substrates remain in the linear phase of the uptake time-course at the time point at which we performed the assays (30 min). We include these new data in Supplementary Fig. 4b, e-f.

10. 349. Hypoargininaemia is an interesting phenomenon, however, this is not well connected to the text, especially as the phenomenon seems attributed to host arginase activity.

We have de-emphasised this in the text, which now reads (Lines 414-415 of the revised manuscript):

“*Plasmodium* NPT1 conceivably plays a role in *Plasmodium*-induced

hypoargininaemia.”

11. General 1: The data do not contain any information on the transport mechanism e.g. driving force, co-transported ions, etc. When working with oocytes two-electrode voltage clamp analyses are obvious and could provide these data. In case substrate-induced currents are too small to perform such analyses, the authors should at least mention this. Also export can be easily measured by injecting radiolabeled amino acids into the oocytes. Though export may not be physiologically relevant, these results would allow drawing additional conclusions on the mechanism of transport. As this is a member of a novel protein family, this would be valuable information that would strengthen the biochemical characterization of NPT1.

As suggested by the reviewer (and editor), we have now performed an extensive mechanistic characterization of *TgNPT1*, using a combination of [¹⁴C]Arg-uptake and two-electrode voltage clamp analyses. These studies (highlighted in Fig. 4 and Supplementary Fig. 5) demonstrate that *TgNPT1* is a Na⁺, Cl⁻, K⁺, Mg²⁺ and Ca²⁺-independent, pH-dependent transporter. Furthermore, we demonstrate that *TgNPT1*-mediated Arg transport is electrogenic, and that *TgNPT1*-mediated, Arg-induced current is Na⁺, Cl⁻, K⁺, Mg²⁺ and Ca²⁺-independent, and pH dependent.

We include the following text in the body of the manuscript (Lines 198-236 of the revised manuscript):

“The transport of arginine via *TgNPT1* is ion-independent, pH-sensitive, and electrogenic. To elucidate the mechanism of arginine transport by *TgNPT1*, we measured the accumulation of [¹⁴C]Arg into *TgNPT1*-expressing oocytes in media in which different ions were removed. Removal of Na⁺, Cl⁻, K⁺, Mg²⁺ or Ca²⁺ had no effect on *TgNPT1*-mediated [¹⁴C]Arg uptake measured over 30 min (Fig. 4a). The pH-sensitivity of *TgNPT1*-mediated [¹⁴C]Arg transport was measured over a pH range of 5-9. [¹⁴C]Arg uptake into *TgNPT1*-expressing oocytes was unaffected by pH over the pH range 5-8, but decreased significantly when the pH was increased above 8 (Fig. 4b).

The electrogenicity of arginine transport by *TgNPT1* was investigated using the electrophysiological two-electrode voltage-clamp technique²¹. Addition of 5 mM arginine to oocytes expressing *TgNPT1* resulted an inward current that reached a maximum of 22 ± 5 nA (mean ± SEM, n=20; Fig. 4c, black trace). The maximum peak current was followed by a spontaneous relaxation (Fig. 4c), reminiscent of the behaviour of the arginine-induced current observed in oocytes expressing the human cationic amino acid transporter hCAT-2a²². Addition of arginine to uninjected oocytes did not induce a current (Fig. 4c, grey trace). The data are consistent with *TgNPT1* mediating the electrogenic uptake of arginine.

The dependence of the arginine-induced current on the ionic composition of the medium was tested by measuring the current in oocytes exposed to media from which Na⁺, Cl⁻, K⁺, Mg²⁺ or Ca²⁺ was absent. In each of the media tested, the arginine-induced current was similar (Fig. 4d), consistent with electrogenic *TgNPT1*-mediated arginine uptake occurring via an ion-independent mechanism. The dependence of the arginine-induced current on pH was tested by measuring the current in media of varying pH. The amplitude of the arginine-induced inward current

was largely insensitive to pH in the pH range 5-8 ($P > 0.05$, $n=8$), but was significantly decreased at pH 9 ($P < 0.01$, $n=8$; Fig. 4e). This mirrored the effects of pH on [^{14}C]Arg uptake (Fig. 4b).

Together, these data demonstrate that TgNPT1 mediates the electrogenic transport of arginine via a mechanism that is independent of Na^+ , Cl^- , K^+ , Mg^{2+} and Ca^{2+} , and which is sensitive to pH at pH values >8 . Whether the electrogenicity arises solely from the transport of the cationic amino acid, or whether it might involve the transport of H^+ is unclear. The observed maximum transport rate derived from the uptake of [^{14}C]Arg equates to a current of 6 nA (1 nA = 36 pmol charges/h); this is lower than the currents measured under voltage clamp conditions, and might be explained by a contribution of H^+ to the observed currents.”

We also describe how we undertook these experiments in the Materials and Methods (Lines 524-527 and Lines 543-558 of the revised manuscript).

12. General 2: For readers not in the field, it would be interesting to get a wider perspective and discussion.

For example:

- How does the transport selectivity, transport mechanism etc. compare with arginine/lysine transporters in other organisms like human, yeast, other protozoan parasites, etc.?

To address this point, we have included the following paragraph in the text (Lines 355-361 of the revised manuscript):

“Arginine uptake in mammalian cells is mediated by broad-specificity cationic amino acid transporters²⁶. By contrast, arginine uptake in *Leishmania* parasites, which are intracellular for a portion of their life cycle, is mediated by a highly selective arginine transporter³⁰, much like TgNPT1-mediated arginine uptake in *T. gondii*. Intracellular parasites are in competition with their host cells for available arginine, perhaps necessitating their having a selective, high affinity arginine transporter to scavenge sufficient amounts of this amino acid for parasite survival.”

- Though NPTs are specific for Apicomplexa, homologs of amino acid transporters from other kingdoms seem to be present in Apicomplexa genomes and may, in addition to other NPTs, contribute to amino acid uptake.

We have incorporated the following text into the final paragraph of the Results and Discussion (Line 441-443 of the revised manuscript):

“Apicomplexans lack SLC7 homologues, but harbour homologues of AAAP family transporters¹², although the functions of these remain to be elucidated.”

13. Minor points

62-72. This could be considerably shortened e.g. by combining the findings in both parasite species, rather than doing this sequentially and by removing parts of the sentences that are not adding e.g. "we go on to show that".

We have modified the section discussing *PbNPT1* to read (Lines 66-68 of the revised manuscript):

“*PbNPT1*, a malaria parasite homologue of *TgNPT1*, functions as a cationic amino acid transporter, implicating cationic amino acid scavenging in gametocyte biology.”

14. 92-96. Molecular mass (Western blot).

"Western blotting revealed that *TgNPT1*-HA has a molecular mass of ~40 kDa, below the calculated mass of 58 kDa (Fig. 1a). The reason for this discrepancy is unclear." Please rephrase: The reason for this discrepancy was not further investigated.

We have modified the text as suggested.

As mentioned, having an apparent smaller molecular mass on SDS PAGE is a common feature of membrane proteins, but cleavage can not be excluded. Whether N-terminal cleavage occurs, could be assessed e.g. by adding a tag at the N-terminus. The information that the author tried adding an N-terminal tag, but could not get respective clones, can only be found in the cloning procedure given in the Suppl. M&M 144. This information could be added to the main text to clarify why this could not be assessed.

We now include the following in the main text (Lines 95-97 of the revised manuscript):

“We attempted to integrate a 5' epitope tag into the *TgNPT1* locus, but were unable to recover genetically modified parasites, precluding further analysis of N-terminal processing.”

15. 97. ...*TgNPT1*-HA localizes to the periphery of the parasite, as well as to some internal structures (Fig.1b).

Please provide information on potential identity of structures (partially also seen in the *SAG1* control) and/or physiological role of transporters in internal compartments.

We did not pursue the internal labeling any further. As the reviewer notes, we also see some internal labelling with the *SAG1* plasma membrane marker, and some of this overlaps with *TgNPT1*. We suspect that this could represent trafficking vesicles and structures that contain proteins destined for the plasma membrane, but without markers for these structures, we cannot be sure.

16. 109. Please rephrase: revealed there to be no detectable.... → revealed no detectable

We have modified the text as suggested.

17. 180, Fig 3A and 6B. Transport in the absence of competitors should be labelled as control, i.e. in the absence of unlabelled amino acids (like in Fig 4b), and not as

"carrier-free".

We have modified this as suggested (see Figs. 3b and 8b, the new versions of the original Figs. 3a and 6b).

18. 303-339, PbNPT1. After presenting the results of *T. gondii*, essentially the same experiments are repeated with *P. berghei*. These results are certainly essential. However, the authors should considerably shorten the *P. berghei* chapter. i.e. the descriptions of the experiments should be more concise. Details can be included in figure legends.

We have shortened this section as follows:

- We have removed the section that reads: "... confirmed its plasma membrane localization (Supplementary Fig. 4d)".

- We have removed the section that reads: "Gene deletion was confirmed by PCR analysis". We now cite Supplementary Fig. 2g-h in the previous sentence.

The remainder of this section describes the results, which we believe must remain in the main body of the text.

19. 330. Also include the published finding that PbNPT1 is localized at the plasma membrane.

We include the following text in the introduction (Lines 57-58) of the revised manuscript):

"*PbNPT1* was shown previously to be a plasma membrane protein that is essential for the development of sexual-stage gametocytes¹⁴."

20. 335. "functionally isolated from their host erythrocyte"... Please rephrase.

We have rephrased this (Lines 393-395):

"... (isolated from their host erythrocytes by saponin permeabilisation of the host cell and parasitophorous vacuole membranes) ..."

21. 440. "suggested that methionine could inhibit". Rephrase → suggested that several amino acids including methionine could inhibit. Can the authors exclude that adding methionine inhibits endogenous transport systems or simply reduced unspecific binding?

The reviewer is correct that we cannot exclude the latter scenario. We have rephrased (Lines 527-529):

"Preliminary experiments indicated that several amino acids, including methionine, either inhibited the uptake of radiolabelled arginine by uninjected

oocytes, or inhibited the non-specific binding of radiolabelled arginine to oocytes (Fig. 3a).”

22. 630. immunofluorescence reveals co-localization. Rephrase: reveals partial co-localization

We have rephrased this as suggested.

23. Please replace lab slang in Supp. Information e.g.

39, and several other lines. "predicted to give a band" → predicted to amplify a fragment

147. "plates were grown" → plates were incubated or cells were grown.

We have modified the text as suggested by the reviewer. Namely, we have changed “band” to “PCR product”, and have changed “plates were grown” to “plates were incubated”

24. Was in any experiment NPT1-mediated transport in the absence of a HA tag compared with NPT1-HA that is generally used? Not sure whether I missed this.

We did not do this experiment. Complementation of *TgNPT1*-depleted parasites with HA-tagged *TgNPT1* restores growth (Fig. 1e), and complementation of $\Delta npt1$ parasites with constitutively-expressed HA-tagged *TgNPT1* ($\Delta npt1/tubNPT1$) restores both arginine uptake (Fig. 5a) and parasite virulence (Fig. 7). These data are consistent with the HA tag not interfering with protein function.

25. General comment: It is not common to do competition experiments at substrate concentrations so far below the K_m of the transporter and use competing substrates at >3000-fold excess. Competition experiments at V_{max} or around K_m are preferred for these analyses. As both background and NPT1-mediated uptake seem to increase linearly at concentrations below K_m , I can not see any reason why the experiments were not performed using concentrations in the physiological range. As background stays reasonably low at these concentrations, I do not expect changes in the outcome. Nevertheless, changing this in future experiments is highly recommended.

We thank the reviewer for this suggestion. In the case of *TgNPT1*, the demonstrated inability of other amino acids to inhibit arginine uptake at such large excess strongly supports that this is a selective arginine transporter.

Reviewer #3 (Remarks to the Author):

Our understanding and exploitation of solute transporters in apicomplexan parasites has been hindered by a lack of functional information about these important proteins. Putative transporters that show little or no homology to previously characterised transporters make attempts to identify and demonstrate function extremely difficult. Thus, the author's efforts here are worthy of some praise and should be of interest to

the readership of Nature Communications. The findings are novel and are supported by the data, which are generally of a high standard, contain correct controls and are analysed appropriately (including statistically). However, I do have a number of suggestions that the authors might consider to improve the paper.

1. Abstract. Line 28. Maybe "characterises" rather than "identifies" since a number of putative amino acid transporters have been identified previously.

We have changed this as suggested.

2. Introduction. Line 41. I disagree with this sentence as characterised transporters also have equivalents in plants and fungi and in some cases are absent in mammals. For example, the Plasmodium and Toxoplasma calcium hydrogen exchangers have been characterised, which do not have orthologues in mammalian cells (PMID: 23468629), and the functionally characterised Plasmodium vacuolar iron transporter also lacks mammalian orthologues (PMID: 26786069).

We agree with the reviewer. The point we are trying to make here is that the substrates of most 'characterized' transporters were predicted based on homology to transporters studied in other organisms. We have rephrased the text (Lines 40-42):

"... and most of those that have been characterized are homologues of equivalent transporters in other organisms, such as mammals, yeast and plants".

3. General. Please could the authors comment on the physiological concentrations of cationic amino acids that these parasites are likely to encounter?

There is relatively little information about this, probably because it is difficult to measure intracellular amino acid levels *in vivo*. We cite one study (Arnal et al, 1995) that demonstrates a correlation between intra- and extra-cellular arginine concentrations, and a second study (Anderson et al, 1985) that notes mouse plasma arginine levels are less than lysine levels. In the revised manuscript, we cite an additional paper (Bergström et al, 1974) that reported [Arg] and [Lys] in human plasma and muscle tissue. They found that the ratio of the two amino acids were equivalent in both plasma and muscle tissue (0.44 in both cases). This is consistent with our hypothesis that *T. gondii* parasites encounter a low Arg:Lys ratio intracellularly. Furthermore, our finding that $\Delta npt1$ parasites are avirulent is consistent with the intracellular Arg:Lys ratio being low *in vivo*. A challenge for the future will be to develop approaches for measuring the concentrations of these amino acids in (infected) host cells. We have modified the text to include mention of the Bergström paper as follows (Lines 326-329 of the revised manuscript):

"An older study noted that the ratio of [Arg]:[Lys] in human plasma is 0.44, and that the ratio in muscle tissue is 0.44²⁸, suggesting a close similarity between the [Arg]:[Lys] ratio in the extracellular environment and that in the cell cytosol."

4. General. I may have missed the section but it's not always clear which statistical test has been used to derive P values, please can this be clarified?

We undertook ANOVA and Student's T test analyses depending on the nature of the

data, as we now specify in the methods (Lines 616-618). We have clarified which we used in the figure legends.

5. Results. Line 117. Here it is stated that 30 mins was used for measuring uptake (and this follows through to PbNPT1 oocyte studies). Why was this time point chosen? Were time courses undertaken? Is uptake linear over this period? As I'm sure the authors are aware, this is important to establish when undertaking concentration-dependence experiments (at both high and low concentrations of unlabelled solute).

See response to point 9 by reviewer 2.

6. Results. Section from Line 171. Did the authors consider the use of L-arginine analogues to probe the selectivity of the transporter further?

We have not done this. While this is a good idea for the future (for instance as a starting point to measure the nature of arginine binding to the transporter), our purpose in this paper was to measure the role of NPT1 in arginine transport.

7. Results. Section from Line 171. Can the authors comment further on the transport mechanism? Facilitative or secondary active? Were additional experiments undertaken to shed light on this?

See response to point 11 by reviewer 2.

8. Results. Section from Line 196. Can the authors speculate on the importance of the alternate arginine pathway at physiological arginine concentrations, if likely to be different from those used (see point 3 also)?

See our response to point 3. We demonstrate that $\Delta npt1$ parasites are avirulent (Fig. 5), consistent with a minimal role for the alternative arginine transporter in mediating arginine uptake in parasites *in vivo*. We demonstrate that the alternative arginine transporter likely also transports lysine, and it is conceivable that it has an essential role in this process. Identifying the molecular identity of this transporter(s) is a priority for future studies.

9. Results. Line 346. PfNPT1 is mentioned at this point as the likely underlying mechanism for cationic amino acid uptake reported recently in *P. falciparum* by some of the authors of this study. Since this particular transporter was not characterised here perhaps the authors could provide some comment as to the likelihood that the Pb and PfNPT1 have similar function e.g. is there sequence similarity between PbNPT1 and PfNPT1 - the sequence could be added to Supp Fig 1?

As suggested, we have modified the alignment in Supplementary Figure 1 to include PfNPT1. This reveals a high degree of similarity between PbNPT1 and PfNPT1. These proteins are also located in syntenic positions of the respective genomes, strong evidence that PbNPT1 and PfNPT1 are orthologues, and therefore likely have equivalent functions. We have included the following text in the legend of

Supplementary Figure 1:

Note that *Pf*NPT1 and *Pb*NPT1 occur in syntenic positions on the *P. falciparum* and *P. berghei* genomes, strong evidence that *Pb*NPT1 and *Pf*NPT1 are orthologues.

10. Results. Line 370. Same as point 1 above.

We have changed this as suggested.

11. Results. Line 372-374. The authors note that NPTs are they are attractive targets, since they lack homologues in humans. This is only one factor of many that make a target attractive. Perhaps the authors could expand on this and/or find an appropriate reference.

We have included the following sentence (Lines 444-447 of the revised manuscript):

We and others have demonstrated that NPT-family proteins are essential for virulence and life cycle progression. These and other essential orphan transporters are therefore attractive targets for therapeutic interventions against apicomplexans.

12. Methods. Line 456. What were the specific activities of radio-labelled arginine and lysine?

We now report the specific activities of radiolabeled arginine, lysine and 2-deoxyglucose in the methods section (Lines 563-564 and Lines 582-583)

13. Methods. Line 471. [¹⁴C]Arg rather than [¹⁴C]arginine for consistency and the same for Lys.

We have changed this as suggested.

14. Figure 4. Although mentioned in the methods section and main text that initial rates of uptake were determined by fitting an exponential function, it is not completely clear that this is the case for the data presented in Figure 4b. Please could this be clarified (e.g. altering the y axis label and figure legend text)? If this is not the case, some of the data could be explained by inhibition of metabolism rather than transport since the parasite are maintained in glucose and thus are likely ATP replete and metabolically active.

In the initial Fig. 4b, we had performed uptake for 30 mins (as an end point assay). Upon the suggestion of the reviewer, we performed additional experiments that demonstrated that this is not in the linear phase of uptake (Supplementary Fig. 6b). As the reviewer notes, we cannot therefore rule out an influence of cellular metabolism on these results. Based on these comments, and comments 4 and 7 by Reviewer 2, we have repeated the experiments shown in Fig. 4a and 4b (now Fig. 5a and 5b). Specifically:

- We have repeated the *T. gondii* uptakes using an oil-stop technique, similar to that which we use for the *P. berghei* uptake experiments. This approach allows us to take more rapid time points, in a more reliable manner, thereby

- giving us a more accurate measure of the initial rate of uptake.
- We have redone the experiment in the original Fig. 4a (now Fig. 5a) including the WT uptake in the presence of lysine, and with the addback control (see response to comment 7 by Reviewer 2).
- We perform a lysine uptake time course experiment to determine the time points at which uptake is in the linear phase, including this as Supplementary Fig. 6b.
- We have redone the experiment in the original Fig. 4b (now Fig. 5b) using a time point (3 mins) at which lysine uptake is still in the linear phase.

These new experiments support our initial conclusions that:

- Loss of NPT1 leads to a significant decrease in the rate of arginine uptake in *T. gondii* parasites.
- Uptake of arginine in $\Delta npt1$ parasites in the presence of unlabeled lysine is significantly decreased (by 75%) compared to WT parasites. This highlights the importance of *TgNPT1* for arginine uptake, and suggests the presence of an alternative arginine-uptake pathway that is modulated by lysine.
- *T. gondii* harbours a general cationic amino acid uptake pathway that is independent of *TgNPT1*.

Line 221: Fig 4a rather than Fig 3a?

We mean to compare our results from Figure 4 (now Fig. 5) with the oocyte data in Figure 3a. So Fig. 3a is correct here.

Also, the transport data in Figure 4 are presented as relative uptake - could the authors provide actual uptake data/calculated initial uptake values for WT and deltaNPT1 parasites to act as a reference point for readers. This could be added to the figure legend.

We have now done this - see our response to point 3 by Reviewer 2.

15. Supp Figure 4. I may have missed this but what was the protein load in this and other westerns? Also Supp Fig 4a: it is very difficult to see if there is even a control image present in the version I have.

We have expanded the description of this experiment in the Materials and Methods. This now reads (Lines 511-515 of the revised manuscript):

“Proteins extracted from 5-10 biotin-treated *TgNPT1*-expressing oocytes, *PbNPT1*-expressing oocytes or uninjected oocytes were affinity purified using streptavidin-conjugated agarose beads. Affinity purified proteins (at a loading equivalent of one oocyte per lane) were separated by SDS-PAGE and subjected to western blotting with anti-HA antibodies.”

Note that there was very low background in the western blot depicted in Supplementary Fig. 4a, hence the inability to see anything in the control lane.

Reviewer #1 (Remarks to the Author)

This manuscript offers a thorough genetic and biochemical dissection of basic amino acid transporters in *Toxoplasma gondii* and *Plasmodium berghei*. As a previous reviewer, I am satisfied by the authors' efforts to address the concerns expressed by the three referees of the initial submission, including additional experiments, and I have no further concerns.

Reviewer #2 (Remarks to the Author)

The authors added a number of additional experiments and addressed all comments raised in my previous review. It is a nice piece of work !

Reviewer #3 (Remarks to the Author)

The authors have addressed previous comments to my satisfaction, for which I am grateful, and I hope that they agree that the manuscript is better for this.